# ERp18 regulates activation of ATF6α during unfolded protein response

Ojore BV Oka[1] (ID), Marcel van Lith[1], Jana Rudolf[2], Wanida Tungkum[1], Marie Anne Pringle[1] & Neil J Bulleid[1,*] (ID)

## Abstract

Activation of the ATF6α signaling pathway is initiated by trafficking of ATF6α from the ER to the Golgi apparatus. Its subsequent proteolysis releases a transcription factor that translocates to the nucleus causing downstream gene activation. How ER retention, Golgi trafficking, and proteolysis of ATF6α are regulated and whether additional protein partners are required for its localization and processing remain unresolved. Here, we show that ER-resident oxidoreductase ERp18 associates with ATF6α following ER stress and plays a key role in both trafficking and activation of ATF6α. We find that ERp18 depletion attenuates the ATF6α stress response. Paradoxically, ER stress accelerates trafficking of ATF6α to the Golgi in ERp18-depleted cells. However, the translocated ATF6α becomes aberrantly processed preventing release of the soluble transcription factor. Hence, we demonstrate that ERp18 monitors ATF6α ER quality control to ensure optimal processing following trafficking to the Golgi.

Keywords ATF6α; ER stress; ERp18; protein trafficking; unfolded protein response

Subject Categories Membrane & Intracellular Transport; Protein Biosynthesis & Quality Control

The EMBO Journal (2019) 38: e100990

See also: **D Fass** (August 2019)

## Introduction

The cellular response to the presence of misfolded proteins within the endoplasmic reticulum (ER) is collectively termed the unfolded protein response (UPR; Ron & Walter, 2007). This stress response involves both translational and transcriptional programs coordinated mainly by three transmembrane transducers localized to the ER: Ire1α, PERK, and ATF6α (Walter & Ron, 2011). The translational program activated by PERK leads to the attenuation of translation following phosphorylation of eIF2α (Lu *et al*, 2004), whereas

activation of Ire1α leads to the processing of unspliced XBP1 mRNA. This processing generates spliced transcripts that are translated to a transcription factor, which upregulates the expression of proteins involved in ER protein folding, ER-associated degradation (ERAD), and lipid biogenesis. Ire1α activity also leads to the degradation of specific mRNAs (Hollien *et al*, 2009). All transducers activate transcriptional programs that lead to the upregulation of genes coding for proteins important for ER expansion, protein folding, and protein degradation (Travers *et al*, 2000; Okada *et al*, 2002; Lu *et al*, 2004). The consequence of these programs is either recovery from the cell stress or, if the stress persists, cell death (Walter & Ron, 2011).

ATF6 exists as two isoforms, α and β (Haze *et al*, 2001), with ATF6α being predominant (Thuerauf *et al*, 2004) and the focus of this study. ATF6α is normally resident within the ER; however, after ER stress, it traffics to the Golgi apparatus. Here, it is cleaved sequentially by site-1 protease (S1P) and site-2 protease (S2P) to release a soluble bZIP transcription factor, ATF6-N (Ye *et al*, 2000). ATF6-N translocates to the nucleus where it binds to ER stress-responsive elements (ERSEs) in the promoters of UPR genes to bring about its downstream effects (Yamamoto *et al*, 2004; Adachi *et al*, 2008). Regulation of activation is afforded by ATF6α retention in the ER and by S1P/S2P being located in the Golgi. Localization of ATF6α is critical to the process of activation and is thought to involve binding partners that release the ER retention of ATF6α (Shen *et al*, 2002; Higa *et al*, 2014) or bind and recruit ATF6α to COPII-coated vesicles following stress (Lynch *et al*, 2012). It is also thought that the oligomeric state of ATF6α is important in regulating trafficking (Nadanaka *et al*, 2007). The ATF6α lumenal domain contains two cysteine residues that can form either intra- or interchain disulfides. ATF6α in unstressed cells exists as a mixture of disulfide-linked oligomers and monomers. Upon stress, the interchain and intrachain disulfides are reduced giving rise to the suggestion that only the reduced monomeric form exits the ER to the Golgi (Nadanaka *et al*, 2007). In addition, there is evidence that the glycosylation status of ATF6α can influence the trafficking of ATF6α as the hypoglycosylated protein is transported more rapidly than glycosylated ATF6α (Hong *et al*, 2004a).

The lumenal domain of ATF6α plays a critical role in regulating ATF6α trafficking. It is both necessary and sufficient to allow ER retention and release following stress (Chen *et al*, 2002; Sato *et al*,

1 Institute of Molecular, Cell and Systems Biology, College of Medical Veterinary and Life Sciences, University of Glasgow, Glasgow, UK
2 Inserm U1035/BMGIC, University of Bordeaux, Bordeaux, France
*Corresponding author. Tel: +44 141 330 3870; Fax: +44 141 330 5481; E-mail: neil.bulleid@glasgow.ac.uk

2011), suggesting that the factors or signals influencing cellular localization reside within this domain. Distinct regions confer binding to the ER-localized Hsp70 homologue BiP and act as Golgi-localization sequences (Shen *et al*, 2002). Binding of BiP to these regions could mask localization signals, which only become accessible after BiP dissociation (Shen & Prywes, 2005). What regulates BiP binding is not known, but competition for binding to misfolded proteins could occur (Schindler & Schekman, 2009). However, BiP binding to ATF6α is relatively stable and does not appear to be regulated by ATP arguing against a simple competition mechanism (Shen *et al*, 2005). Other proteins shown to bind to ATF6α that regulate ER-to-Golgi trafficking include thrombospondin 4, which promotes its nuclear shuttling in cardiomyocytes (Lynch *et al*, 2012), and PDIR, an ER oxidoreductase which modulates packaging of ATF6α into COPII-coated vesicles (Higa *et al*, 2014). While we know the identity of some of the proteins that associate with ATF6α that may prevent or promote trafficking, what influences their association/dissociation remains unknown.

To evaluate the regulation of trafficking of ATF6α, we took an unbiased approach to identify ER proteins interacting with its lumenal domain in normal and stressed conditions. From the list of interactors, we identify an ER oxidoreductase, ERp18, that associates with ATF6α only after ER stress. Further characterization of the role of ERp18 in ATF6α activation reveals that it regulates ER exit and in doing so optimizes cleavage by S1P/S2P in the Golgi. These results uncover an unexpected layer of control over the activation of ATF6α at the level of Golgi processing with only a specific form of ATF6α being processed correctly to release soluble transcription factor.

# Results

### ERp18 only associates with ATF6 following ER stress

To understand more fully the process of ATF6α activation, we created a stable cell line expressing epitope-tagged ATF6α (Fig 1A) and confirmed its expression and ER location by immunofluorescence (Fig 1B). We evaluated the proteins interacting with ATF6α before or after ER stress generated by thapsigargin-dependent calcium depletion (Li *et al*, 1992). To ensure we captured weakly interacting proteins and to minimize non-specific binders, we added an amine-specific thiol-cleavable cross-linking agent, immunoisolated ATF6α complexes, and eluted interacting proteins with the reducing agent DTT. Eluted proteins were digested with trypsin and the peptides identified by mass spectrometry. The experiment was carried out three times, and the proteins consistently observed are listed along with the % coverage and exponentially modified protein abundance index (emPAI) scores from a representative experiment (Table 1). None of these proteins were present in control samples from untransfected HEK293 cells. The formation of ATF6α complexes with several of the proteins identified was confirmed by their immunoisolation with the V5 antibody and detection by Western blot (Fig 1C). RCN1 was transiently transfected into cells for this experiment as our RCN1 antibody did not detect endogenous protein in HEK293 cells. The interactions were specific to ATF6α as the signals were much lower or not present in its absence. The presence of Grp170 in the immunoisolate from untransfected cells is likely due to non-specific binding to the agarose beads.

The list of ATF6α interacting proteins is dominated by ER-resident folding factors including chaperone proteins and folding enzymes such as BiP, Grp94, and members of the protein disulfide isomerase (PDI) family. In addition, proteins involved in glycoprotein quality control such as calreticulin and glucosidase II were identified along with members of the CREC family of calcium-binding proteins, reticulocalbin (1 and 2) and calumenin (Honore, 2009). Some changes to the ATF6α interactome occurred following ER stress which include the dissociation of the CREC family members as well as Grp170 and glucosidase II. This result is expected as the ER stress was caused by calcium depletion and all of these proteins are calcium-binding proteins which could regulate their interaction with ATF6α. Absent from the list of interactors was PDIR and thrombospondin 4, which have previously been shown to interact with ATF6α (Lynch *et al*, 2012; Higa *et al*, 2014). An interaction between endogenous PDIR and ATF6α before and after ER stress was observed by Western blotting after immunoisolation of ATF6α (Fig 1D). Endogenous thrombospondin 4 expression in HEK293 cells was below the level of detection by Western blot, but an interaction with ATF6α was observed before and after ER stress following co-expression of thrombospondin 4 with ATF6α (Fig 1D).

The PDI family member ERp18 was found to associate with ATF6α only after ER stress. This protein has a single thioredoxin domain and has been shown previously to have thiol-disulfide oxidase and isomerase activity as well as protecting cells against prolonged ER stress (Alanen *et al*, 2003; Jeong *et al*, 2008). We confirmed the mass spectrometry result by demonstrating an interaction between ERp18 and ATF6α using an immunoblotting approach (Fig 1D). We detected ERp18 in ATF6α immunoisolated material in unstressed cells, the amount of which increased following ER stress (Fig 1D). Having shown that ERp18 interacts with ATF6α, we decided to evaluate the role of this protein and other PDIs during the activation process.

### ERp18 is catalytically active toward ATF6α during ER stress

We expressed substrate-trapping mutants of various V5-tagged PDI family members in the ATF6α-expressing stable cell line to indicate which PDIs could catalyze disulfide reduction. Mutant PDIs where the active site was converted from CXXC to CXXA were used to trap substrates in covalent complexes with the enzyme (Jessop *et al*, 2007, 2009; Zito *et al*, 2010; Oka *et al*, 2013). The formation of a mixed disulfide between enzyme and substrate indicates that the first step in catalysis can occur (Walker *et al*, 1996). We chose the catalytic PDIs identified as interacting with ATF6α in this study (P5, ERp72, ERp57, PDI, and ERp18) or by others (PDIR; Higa *et al*, 2014) as well as ERp46 and TMX1 which had not been shown previously to interact. When ATF6α was immunoisolated with anti-ATF6α, separated under non-reducing conditions, and immunoblotted with anti-HA, the resulting banding pattern indicated the presence of ATF6α monomers (M), dimers (D), and oligomers (O) as previously described (Nadanaka *et al*, 2007; Fig 2A, lane 1). In addition, clearly defined mixed disulfides were observed between ATF6α and ERp18, ERp57, and PDIR as indicated by the presence of additional bands in the transfected cells that are absent in the untransfected cells (Fig 2A, lanes 2–9, indicated with an arrow). No mixed disulfides were observed with PDI itself, ERp46,

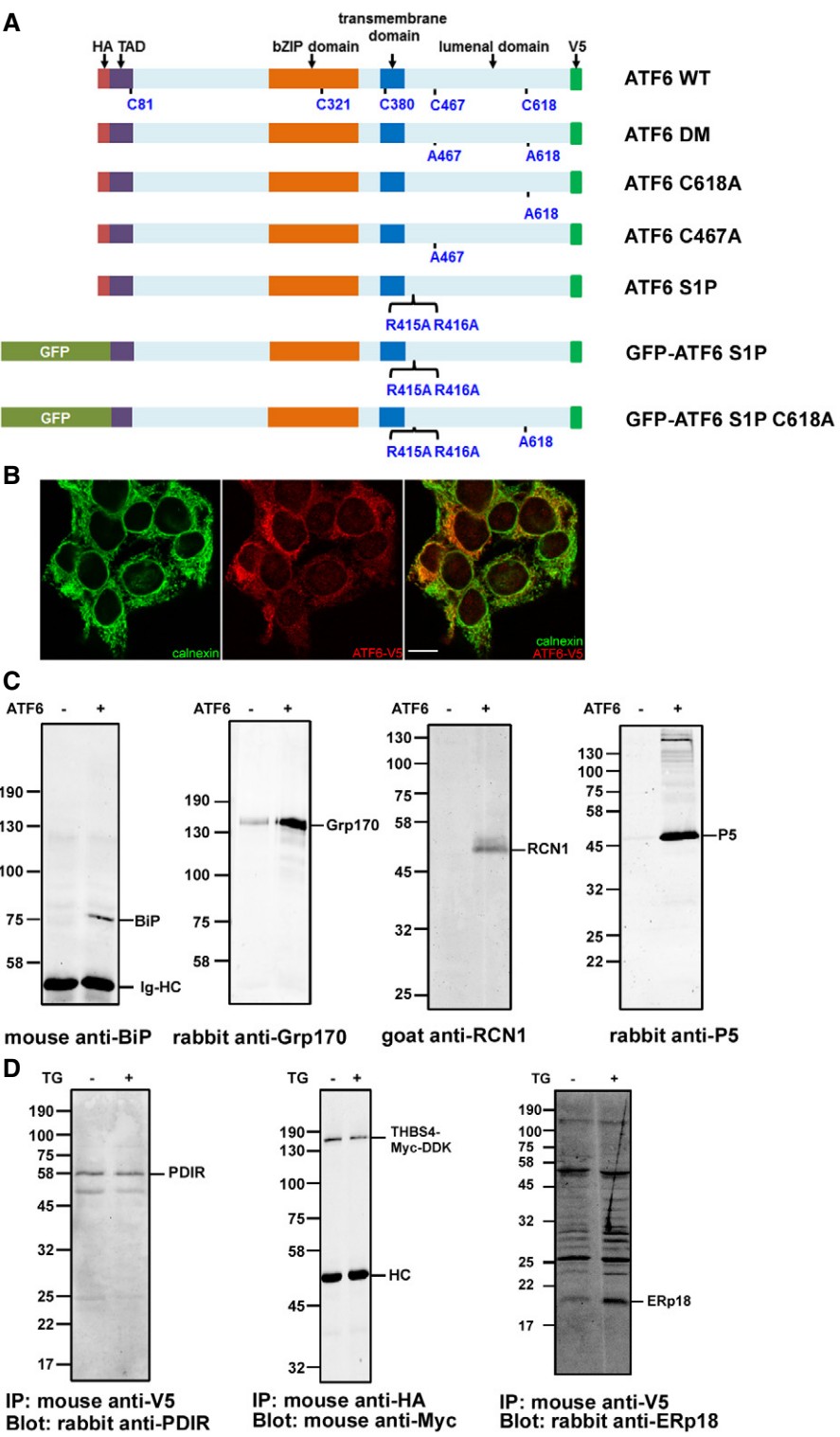

**Figure 1.**

ERp72, P5, or TMX1. As expected, no mixed disulfides with PDIs were observed with a mutant ATF6α (DM) that does not contain any cysteines within its ER lumenal domain (Fig 2B). Our results demonstrate that under non-stressed conditions, a subset of PDIs including ERp18 can act as reductases toward ATF6α.

To determine the consequence of ER stress on the ability of the PDIs to catalyze reduction of ATF6α, we co-expressed PDI substrate-trapping mutants with an S1P cleavage site mutant of ATF6α that is not processed in the Golgi (Ye *et al*, 2000). We expressed ERp18, ERp57, and PDIR individually in the absence or

**Figure 1. Validation of ATF6 partner proteins.**

A   Schematic of the ATF6α constructs used in this study. The N-terminus of ATF6α is tagged with a HA epitope or GFP as indicated. The C-terminus is V5-epitope-tagged. The region N-terminal to the transmembrane domain is localized in the cytosol, whereas the region C-terminal is localized to the ER (lumenal domain). The positions of the cysteine residues and mutations are as indicated as well as the S1P cleavage site mutation (R415A/R416A). TAD: transactivation domain.

B   Immunofluorescence microscopy of HEK293T exogenously expressing ATF6α-V5 tagged. Cells were fixed and stained with anti-calnexin (green) or anti-V5 (red). An overlay of the cells is presented in the third panel. Bar = 10 μm.

C   Whole-cell lysates from HEK293T cells, with or without stable expression of ATF6α (as indicated), were immunoisolated with mouse anti-V5. RCN1 was transiently transfected into both cell lines in the experiment depicted in panel 3. Immunoisolated material was separated by SDS–PAGE under reducing conditions followed by Western blotting to determine the co-isolation of ER proteins with ATF6α. Ig-HC (anti-BiP blot) indicates immunoglobulin heavy chains.

D   Whole-cell lysates from HEK293T cells stably expressing ATF6α were either untreated (−) or treated (+) with 5 μM thapsigargin (TG) for 1 h. THBS4 was transiently transfected into cells in the experiment depicted in panel 2. ATF6α was immunoisolated with either anti-V5 (panels 1 and 3) or anti-HA (panel 2). For the experiment in panel 3, the cells were cross-linked prior to immunoisolation. Samples were separated by SDS–PAGE under reducing conditions followed by Western blots to determine co-isolation of endogenous PDIR, ERp18, or exogenously expressed THBS4 with ATF6α. HC indicates immunoglobulin heavy chain.

Source data are available online for this figure.

**Table 1.   Proteins cross-linked to ATF6 before and after treatment with thapsigargin.**

| Gene | Protein | ATF6 | | ATF6 + Thapsigargin | |
|---|---|---|---|---|---|
| | | % coverage | emPAI | % coverage | emPAI |
| ATF6A | ATF6 | 6 | 0.14 | 6 | 0.21 |
| HSPA5 | BiP | 41 | 2.94 | 42 | 4.12 |
| ENPL | Grp94 | 29 | 0.98 | 26 | 0.58 |
| HYOU1 | Grp170 | 3 | 0.04 | 0 | 0 |
| PDIA6 | P5 | 34 | 1.04 | 36 | 1.26 |
| PDIA4 | ERp72 | 39 | 2.01 | 37 | 0.79 |
| PDIA3 | ERp57 | 31 | 1.19 | 26 | 0.69 |
| PDIA1 | PDI | 14 | 0.18 | 26 | 0.18 |
| ERP29 | ERp29 | 20 | 0.38 | 11 | 0.38 |
| TXNDC12 | ERp18 | 0 | 0 | 13 | 0.28 |
| ERO1A | ero1 | 6 | 0.11 | 6 | 0.11 |
| PRDX4 | peroxiredoxin 4 | 9 | 0.17 | 16 | 0.17 |
| CALR3 | calreticulin | 17 | 0.1 | 0 | 0 |
| RCN1 | reticulocalbin 1 | 10 | 0.13 | 0 | 0 |
| RCN2 | reticulocalbin 2 | 5 | 0.14 | 0 | 0 |
| CALU | calumenin | 10 | 0.14 | 6 | 0.14 |
| PPIB | cyclophilin B | 30 | 0.86 | 32 | 0.86 |
| PRKCSH | glucosidase 2 subunit beta | 7 | 0.08 | 6 | 0 |
| VCP | VCP | 3 | 0.05 | 10 | 0.11 |
| UTS2R | urotensin-2 isoform b | 6 | 0.38 | 6 | 0.38 |

The table shows percent coverage and emPAI values from a representative experiment. ATF6 + Thapsigargin indicates cells treated with 5 μM thapsigargin for 1 h before cross-linking. Experiment was carried out at least three times.

presence of thapsigargin, tunicamycin, or MG132 to provide three alternative mechanisms to induce ER stress: calcium depletion, glycosylation, and proteasome inhibition, respectively. Under non-stressed conditions, very similar patterns of mixed disulfides were observed following immunoisolation of the S1P cleavage site mutant of ATF6α and immunoblotting with anti-HA as seen with the wild-type protein (cf. Figs 2A and 3A, lanes 1, 3, and 5). Under stressed conditions, the mixed disulfides between ERp57 or PDIR and ATF6α were either abolished or greatly diminished irrespective of the ER stressor (Fig 3A–C, lanes 4 and 6). In contrast, of the two

major mixed disulfides between ERp18 and ATF6α, one at ~200 kDa was lost whereas the second at ~150 kDa persisted following ER stress (lane 2). Note the additional products seen following tunicamycin or MG132 treatment indicated with ** that are likely hypoglycosylated ATF6α. In all samples, we also noted a product at approximately 70 kDa that we designated ATF6-P which was prominent when we expressed the S1P mutant of ATF6α. This product will be discussed in more detail below but is likely to be the same as that seen previously when ATF6α was expressed in the presence of an S1P inhibitor (Ye *et al*, 2000; Gallagher *et al*, 2016).

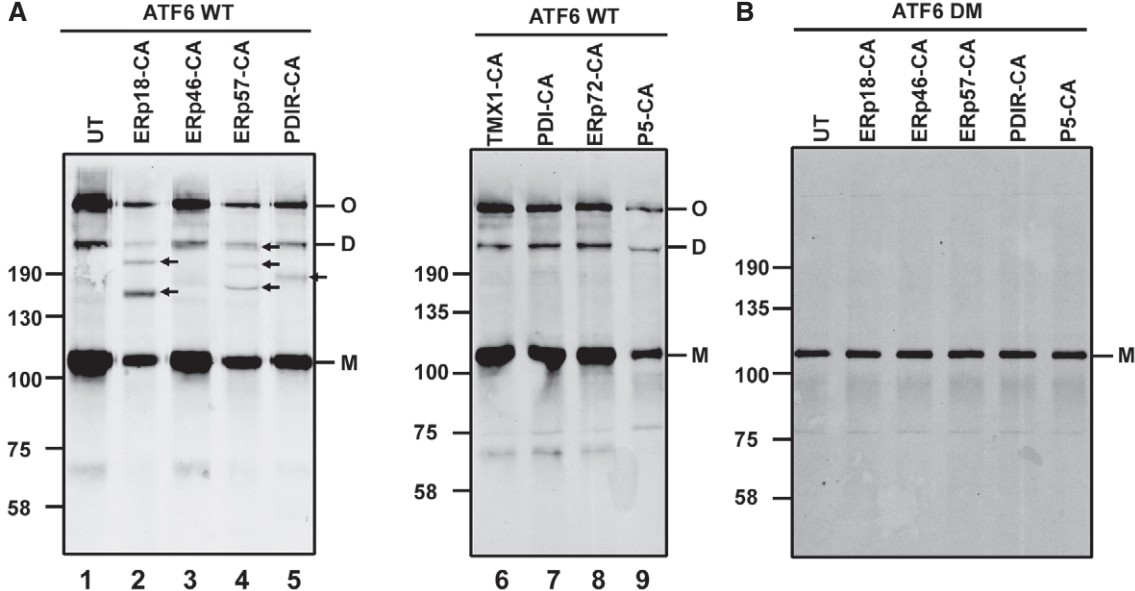

**Figure 2. PDI family substrate-trapping mutants form mixed-disulfide complexes with ATF6.**

A HEK293T cells stably expressing ATF6α were either left untransfected (UT) or transfected with substrate-trapping mutants of PDI oxidoreductases. ATF6α was immunoisolated from cell lysates with mouse anti-ATF6α and separated by SDS–PAGE under non-reducing conditions and ATF6α detected by Western blotting using rabbit anti-HA. Blots confirm mixed-disulfide complexes between ATF6α and ERp18, ERp57, and PDIR indicated with arrows (lanes 2, 4, and 5). M, D, and O refer to ATF6α monomer, dimer, and oligomer, respectively.

B Whole-cell lysates of HEK293T cells expressing an ATF6α mutant containing cysteine-to-alanine mutations in the lumenal domain (ATF6 DM) were transfected with PDI substrate-trapping mutants. ATF6α was immunoisolated with mouse anti-ATF6α and separated by SDS–PAGE under non-reducing conditions and ATF6α detected with rabbit anti-HA.

Source data are available online for this figure.

The presence of each PDI in a mixed disulfide with ATF6α was verified by reprobing the Western blots with antibodies to the PDIs (Fig 3D). These results with the substrate-trapping mutants demonstrate that ERp18 forms mixed disulfides with ATF6α even after ER stress.

The presence of mixed disulfides would indicate that either an intra- or interchain disulfide within the lumenal domain of ATF6α is reduced by the substrate-trapping mutant of ERp18. To investigate this further, we created mutants of ATF6α that had one of the cysteines within the lumenal domain converted to alanine. Single mixed-disulfide species were seen with either cysteine mutant in the absence of ER stress (Fig 4A, lanes 3 and 5), indicating that ERp18 reduces C467 or C618 that has formed an interchain disulfide or has become glutathionylated. Interestingly, the mixed disulfide to the C467A mutant was sensitive to ER stress whereas the mixed disulfide to the C618A mutant was not (lanes 4 and 6). These results indicate that interchain disulfides present in the ATF6α lumenal domain can be reduced by ERp18 during and after ER stress and that the mixed disulfide that persists following stress is via the C467 cysteine.

The mobility of the mixed disulfides was different between the cysteine mutants. A ~130-kDa species was formed with the C467A mutant and a ~150-kDa species with the C618A mutant. The mobility of the interchain disulfide-bonded ATF6α was also different between the two mutants, consistent with previously published data (Nadanaka *et al*, 2007). These mobility differences could reflect different hydrodynamic volumes of the denatured mixed disulfides or be due to additional protein(s) associating with the C618A mutant of ATF6α. To identify additional interacting proteins, we expressed a GFP-tagged S1P cleavage site mutant ATF6α with or without the C618A mutation (Fig 1A). Cells were co-transfected with the substrate-trapping mutant of ERp18, and ATF6α complexes were affinity-purified using GFP-Trap. When the isolated complexes were separated by SDS–PAGE under non-reducing conditions and silver-stained, the presence of both interchain disulfide-bonded ATF6α and the mixed disulfides was observed (Fig 4B). These complexes were analyzed following trypsin digestion and mass spectrometry. No additional protein to ATF6α was present in the slowest migrating band labeled oligomer (O), and only ATF6 and ERp18 were present in the mixed disulfides indicated with arrows (Fig 4B, lanes 2 and 3). The experiment was repeated several times with the same result. We suggest that the differences in mobility seen for both the interchain disulfide-bonded ATF6α and the mixed disulfides are unlikely to be due to additional proteins interacting with the C618A mutant of ATF6α. Hence, the differences in the disulfide-bonding patterns may be due to a difference in hydrodynamic volume after protein denaturation. The two interchain disulfide-bonded species referred to as dimer and oligomer in the wild-type protein are likely to both be dimers which have different electrophoretic mobility due to the disulfide forming between C467 (slower mobility) or C618 (faster mobility). A cartoon summarizing our interpretation of the mixed disulfides between ATF6α and

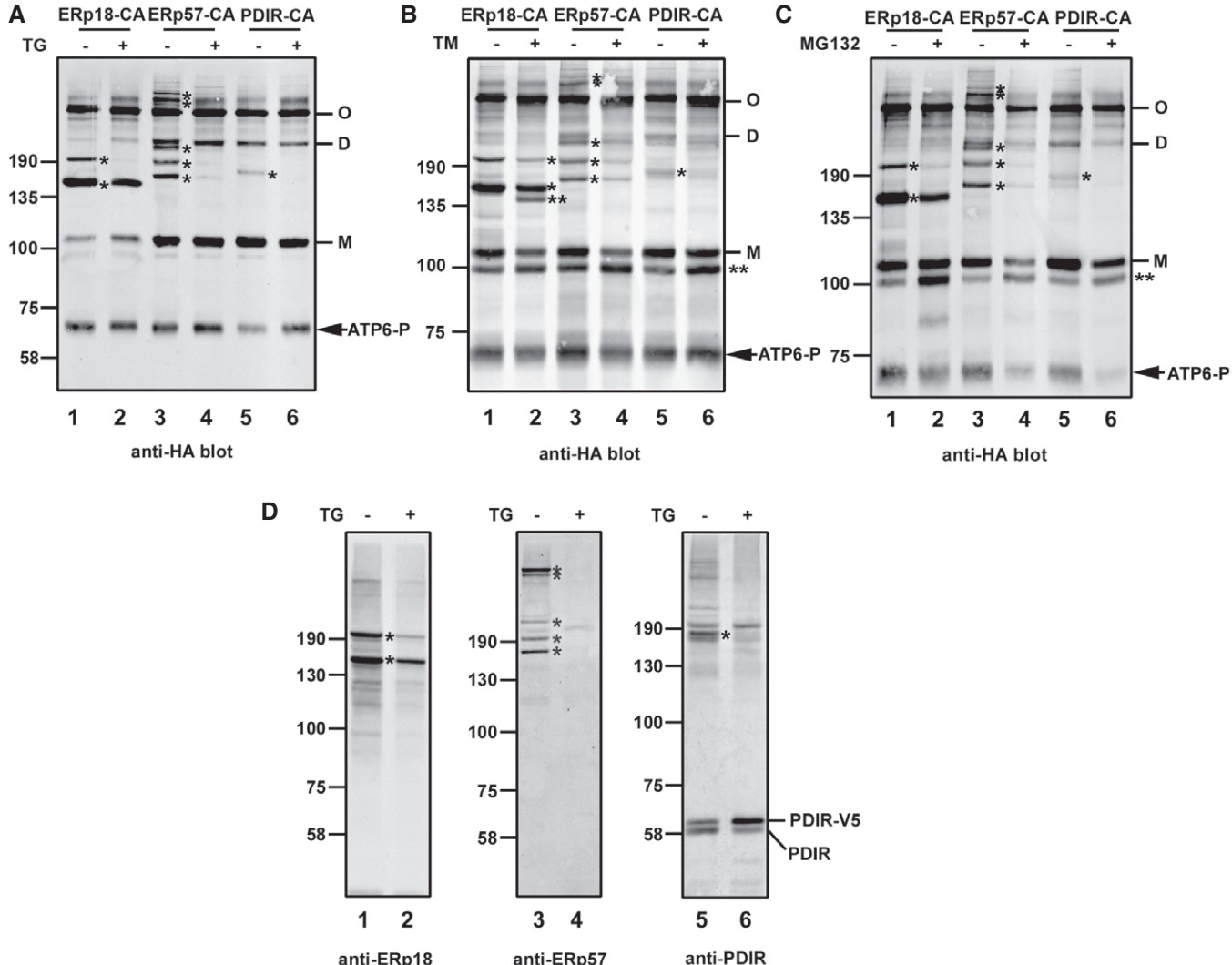

**Figure 3. ER stress modulates ATF6α–PDI mixed-disulfide complexes.**

A–D   HEK293T cells were co-transfected with ATF6α S1P and PDI substrate-trapping mutants and either left untreated (−) or treated (+) for 1 h with 5 μM TG (A), 5 μg/ml tunicamycin (B), or 20 μM MG132 (C) to induce ER stress. ATF6α was immunoisolated from cell lysates with mouse anti-ATF6α and separated by SDS–PAGE under non-reducing conditions and ATF6α detected by Western blotting using rabbit anti-HA. Blots indicate that mixed-disulfide complexes between ATF6α and PDI enzymes are regulated by ER stress. M, D, and O refer to ATF6α monomer, dimer, and oligomer, respectively. Samples from (A) were rerun and probed with anti-sera raised against respective PDI enzymes as indicated (D). * indicates positions of the ATF6α–PDI enzyme mixed-disulfide complexes. ** indicates hypoglycosylated ATF6α. Data shown are representative of three independent experiments.

Source data are available online for this figure.

ERp18 and the potential ATF6α disulfide-bonded dimers is shown in Fig 4C.

**Deletion of ERp18 attenuates the ATF6α stress response**

To determine the function of ERp18 during ATF6α activation, we created an ERp18 KO cell line (Fig 5A). To evaluate whether ERp18 acts during the initial folding of ATF6α, we carried out a pulse-chase experiment (Fig EV1). No difference in the kinetics of appearance of the interchain disulfide-bonded forms of ATF6α was observed between the wild-type and KO cells. This result, together with the observation that ERp18 is recruited to ATF6α following activation,

suggests that its role is not to assist with the initial folding of the lumenal domain.

Next, we evaluated the mRNA levels of known UPR-target genes in the ERp18 KO cell, comparing levels with those in wild-type cells before or after treatment with thapsigargin as an inducer of ER stress. No difference in expression of UPR-target genes was seen in the absence of ER stress (Fig 5B); however, the induction of BiP mRNA was dramatically attenuated in the KO cells following ER stress whereas other UPR-target gene induction was unaffected (Fig 5C). BiP and ERp72 are known targets for ATF6α as well as other pathways, whereas Grp94 is a target for the ATF6α and Ire1 pathways (Lee *et al*, 2003; Adachi *et al*, 2008) and ATF4 and XBP1

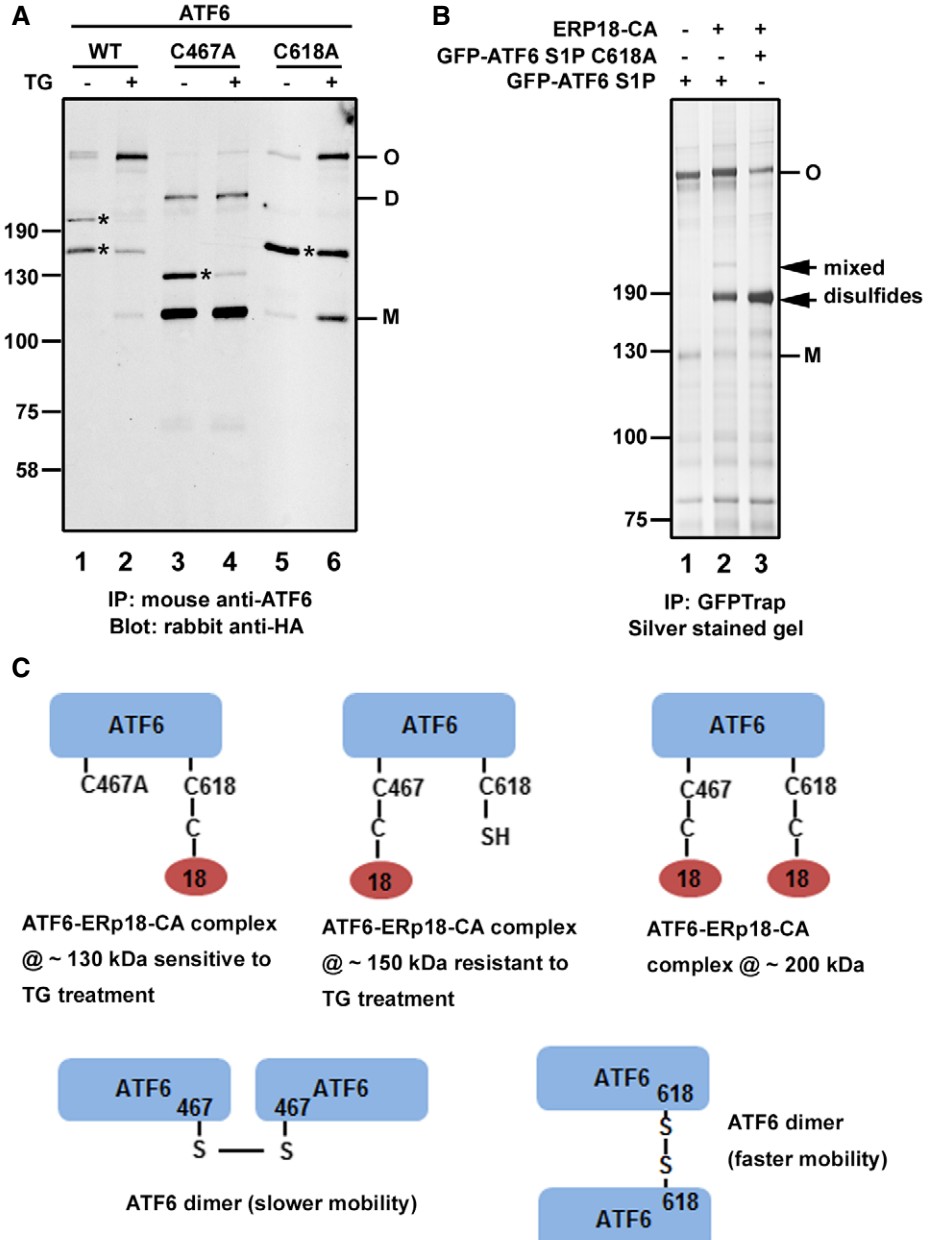

**Figure 4. ATF6α C467 forms a stable mixed-disulfide complex with the ERp18 substrate-trapping mutant.**

A    HEK293T cells were co-transfected with the ERp18 substrate-trapping mutant and the indicated ATF6α constructs. Cell were allowed to recover for 24 h post-transfection before being either left untreated (−) or treated (+) for 1 h with 5 μM TG to induce ER stress. ATF6α was immunoisolated from cell lysates with mouse anti-ATF6α and separated by SDS–PAGE under non-reducing conditions and ATF6α detected by Western blotting using rabbit anti-HA. The blot confirms that ATF6α C618A forms a stable complex with ERp18 trapping mutant following ER stress. M, D, and O refer to ATF6 monomer, dimer, and oligomer, respectively. * indicates positions of the ATF6–ERp18 mixed-disulfide complexes.

B    HEK293T cells were co-transfected with GFP-ATF6α and ERp18 trapping mutant constructs as indicated. GFP-ATF6α complexes were immunoisolated using GFP-Trap. Immunoisolated material was separated by SDS–PAGE under non-reducing conditions and silver-stained to detect ATF6α disulfide-linked complexes. MS analyses confirmed that protein bands indicated by black arrows are a complex comprising ATF6α and ERp18, while bands marked as monomer (M) and oligomer (O) contain ATF6α only.

C    Cartoon depicting the presumed composition of the ATF6α disulfide-linked complexes and dimers indicating their respective apparent molecular weight.

Source data are available online for this figure.

are targets for the PERK pathway (Adamson *et al*, 2016). We also determined the levels of protein expression following UPR induction. We observed an attenuation of BiP protein induction in the ERp18 KO cells as well as the attenuation of induction of ERp72 (Fig 5D–G). These results indicate that the ATF6 pathway in the ERp18 KO cell line is selectively attenuated.

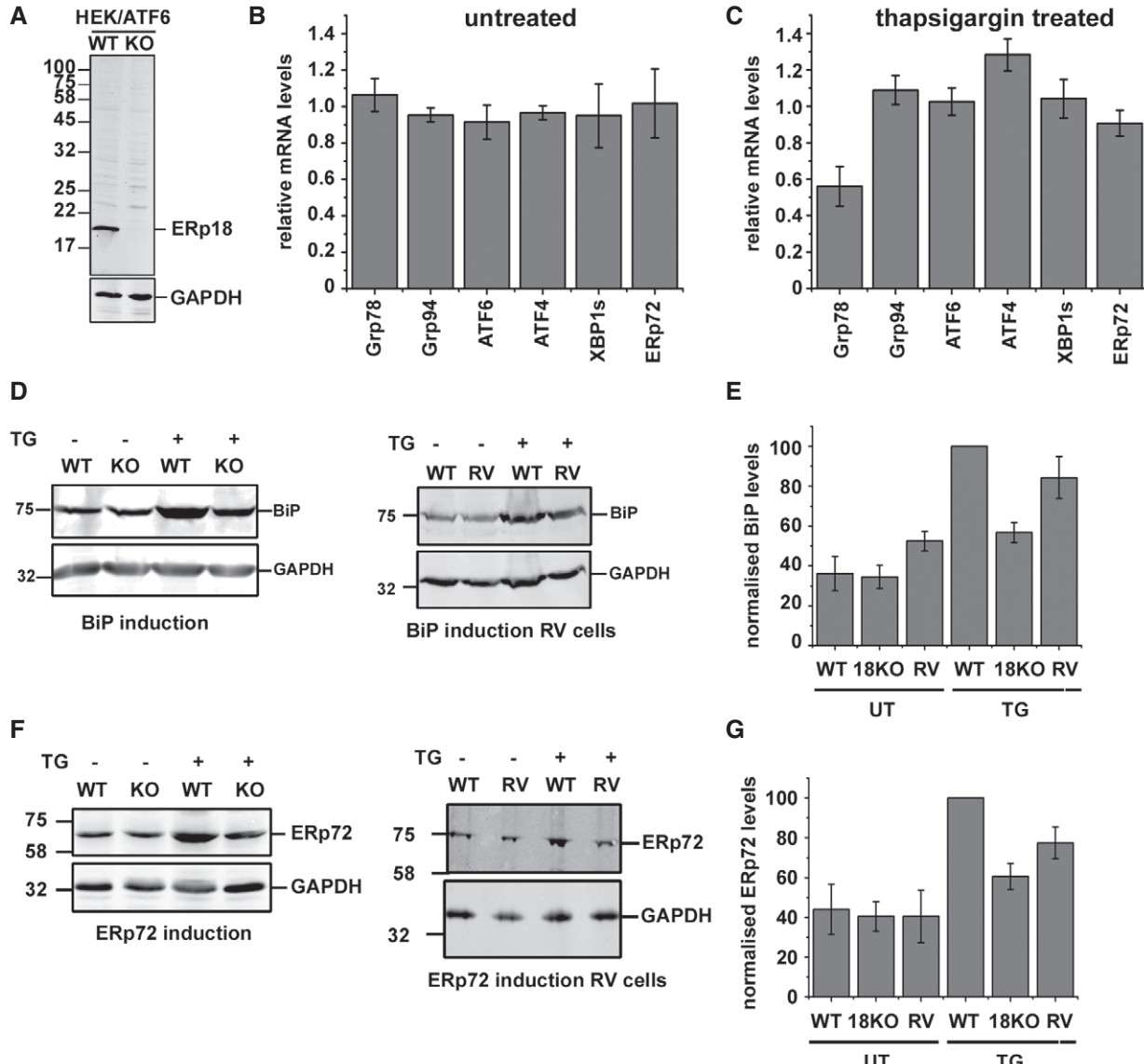

**Figure 5. ERp18 knockout downregulates the transcription and translation of ATF6α target genes during ER stress.**

A   CRISPR/Cas9-based knockout of ERp18 in HEK293T cells expressing ATF6α. ERp18 levels in control (WT) and knockout (KO) cells were detected with an antibody to ERp18.

B   qPCR analysis of UPR-target genes in the absence of ER stress. Wild-type and ERp18 knockout HEK293T cells were untreated prior to isolation of mRNA for qPCR analysis. mRNA levels for Grp78, Grp94, ATF6α, ATF4, XBP1s (spliced form of XBP1), and ERp72 were normalized to GAPDH and then ERp18 KO levels compared to wild type. Error bars represent ± standard deviation for three independent experiments.

C   qPCR analysis of UPR-target genes following ER stress. Wild-type and ERp18 knockout HEK293T cells were treated with 1 μM TG for approximately 16 h prior to isolation of mRNA for qPCR analysis. mRNA levels for Grp78, Grp94, ATF6α, ATF4, XBP1s (spliced form of XBP1), and ERp72 were normalized to GAPDH and then ERp18 KO levels compared to wild-type controls. Error bars represent ± standard deviation for three independent experiments.

D   Levels of BiP protein in wild-type or ERp18 KO cells before and after ER stress induction with thapsigargin (TG). Panel 1 compares wild-type and ERp18 KO cells as indicated. Panel 2 compares ERp18 wild-type cells to ERp18 KO cells that have been transfected with ERp18 (RV).

E   Normalized BiP levels from (D) were quantified. Error bars represent ± standard deviation for three independent experiments.

F   Levels of ERp72 protein in wild-type or ERp18 KO cells before and after ER stress induction with thapsigargin (TG). Panel 1 compares wild-type and ERp18 KO cells as indicated. Panel 2 compares ERp18 wild-type cells to ERp18 KO cells that have been transfected with ERp18 (RV).

G   Normalized ERp72 levels from (F) were quantified. Error bars represent ± standard deviation for three independent experiments.

Source data are available online for this figure.

To verify that the attenuation of the ATF6 pathway was due to the ERp18 KO, we evaluated the consequence of restoring ERp18 expression. For these experiments, we transfected ERp18 into the KO cell line and evaluated the induction of BiP or ERp72 using thapsigargin (Fig 5E and G). Expression of ERp18 reversed the attenuation of BiP or ERp72 induction, demonstrating that the KO phenotype was due to the absence of ERp18 rather than an off-target effect.

   

## The absence of ERp18 results in an additional ATF6α cleavage product

To evaluate further the role of ERp18 in ATF6α activation, we determined the processing events that occur following ER stress comparing the wild-type to the ERp18 KO cells. We separated membrane and nuclear-associated fractions to indicate the location of processed ATF6α (Gallagher *et al*, 2016). Cells were treated for up to 4 h with thapsigargin. In the absence of ER stress, a prominent cleavage product of ATF6α with the same mobility as ATF6-P was

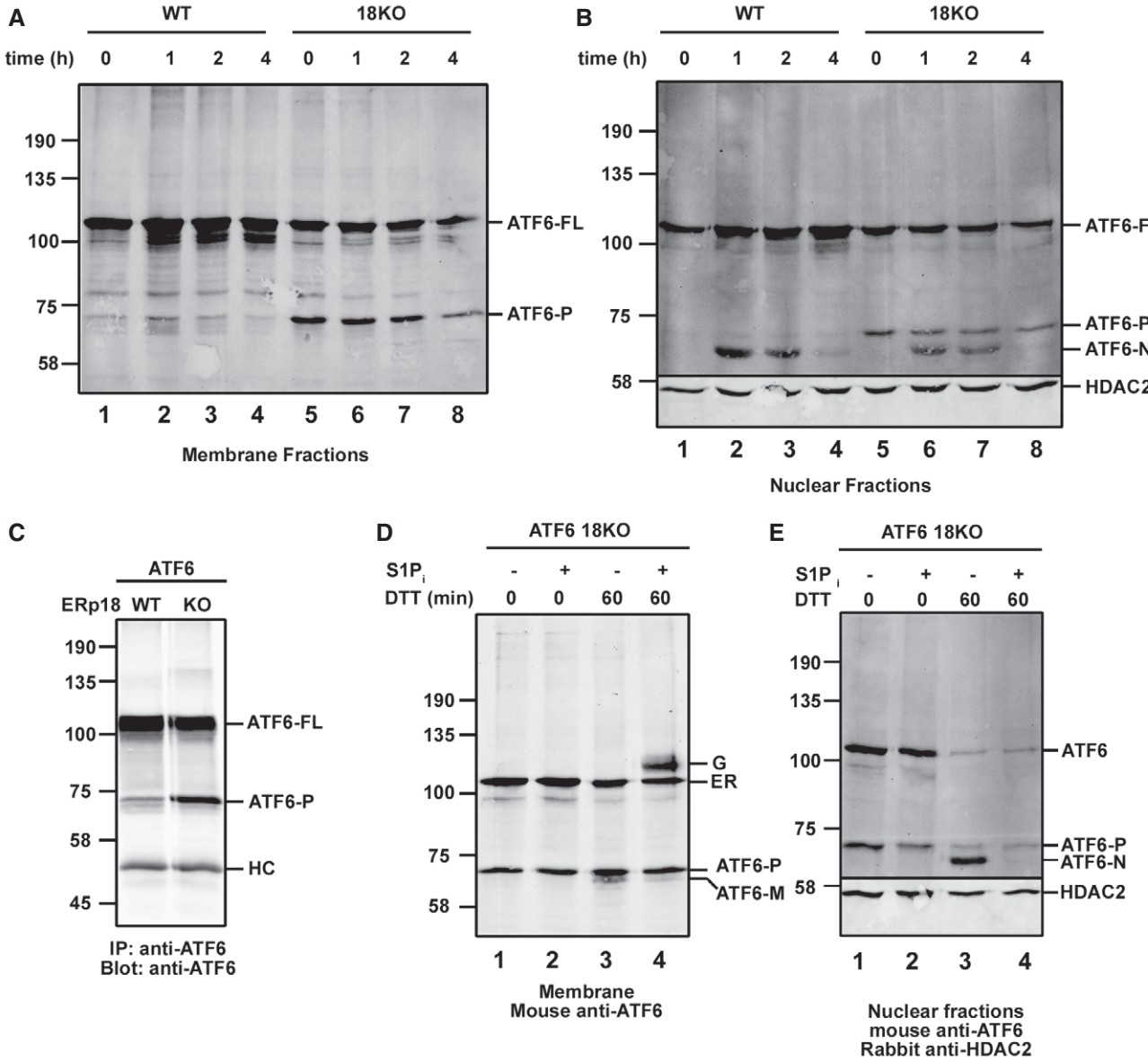

**Figure 6. Deletion of ERp18 regulates cleavage of ATF6α.**

A, B   Wild-type and ERp18 KO cells stably expressing ATF6α were treated with 5 μM TG for the indicated times. Cells were then subject to differential centrifugation to obtain membrane (A) and nuclear fractions (B) which were separated by SDS–PAGE under reducing conditions and analyzed by Western blotting for ATF6α. The positions of full-length ATF6α (ATF6-FL), cleaved membrane-associated ATF6α (ATF6-P), and nuclear-translocated S2P-cleaved ATF6α (ATF6-N) are as indicated. HDAC2 was used as a nuclear marker. Data shown are representative of two independent experiments.

C   ATF6α was immunoisolated from wild-type or ERp18 KO cells expressing ATF6α. Samples were separated by SDS–PAGE, and ATF6α was detected by Western blotting. HC indicates immunoglobulin heavy chains.

D, E   Blots of membrane and nuclear fractions respectively from ERp18 KO cells overexpressing ATF6α, either untreated (−) or treated (+) with 30 μM PF429242 (S1P inhibitor) prior to and following induction of ER stress with 10 mM DTT. The positions of ATF6-P, ATF6-M, and ATF6-N are as indicated. The unprocessed ER-localized ATF6α (designated ER) and the O-linked glycan-modified Golgi-translocated ATF6α (designated G) are indicated. HDAC2 was included as a nuclear marker. The blots in (D and E) confirm that ATF6-P is produced independently of S1P and also the absence of ATF6α processing to ATF6-N in the presence of the S1P inhibitor.

Source data are available online for this figure.

observed in the KO cells (Fig 6A and B, lane 5). This cleavage product was also seen in wild-type cells but to a much lesser extent (lane 1). When an immunoisolation of ATF6α was carried out followed by a Western blot, the presence of ATF6-P was more clearly observed in wild-type cells (Fig 6C). ATF6-P was also observed when the S1P cleavage mutant of ATF6α was expressed (Fig 3A–C). Following thapsigargin treatment, additional faster-migrating products were observed consistent with S1P and then S2P cleavage (Fig 6A, lane 2; Fig 6B, lanes 2, 3, 6, and 7). The fastest migrating species was only observed in the nuclear fraction, indicating that it corresponds to ATF6-N. In the wild-type cells, the S1P product (ATF6-M), which migrates between ATF6-P and ATF6-N, is maximally observed in the membrane fraction at 1 h with the ATF6-N product appearing at the same time in the nuclear fraction. Both cleavage products are diminished after 4 h consistent with previous observations that ATF6-N is rapidly degraded following activation (Hong et al, 2004b). ATF6-P in the KO cells persisted in the membrane fraction throughout the time course. These results indicate that a consequence of ERp18 KO is the cleavage of ATF6α in the absence of stress.

The cleavage of ATF6α in the absence of ER stress could be due to its trafficking to the Golgi and subsequent cleavage by S1P or an alternative protease. We saw no reduction in intensity of ATF6-P in the membrane fraction after cells were incubated in the presence of an S1P inhibitor (Fig 6D, lanes 1 and 2). The S1P inhibitor was functional as combined treatment with DTT resulted in the buildup of a slower migrating full-length ATF6α that has previously been reported to be the result of O-linked glycan modification in the Golgi

(lane 4; Ye et al, 2000; Shen et al, 2002). In addition, treatment with the S1P inhibitor blocked the appearance of ATF6-N in the nuclear fraction following DTT activation (Fig 6E, lanes 3 and 4). These results suggest that the proteolytic processing of ATF6α observed in ERp18 KO cells in the absence of ER stress is not caused by the S1P.

To determine where within the cell the cleavage of ATF6α to form ATF6-P takes place, we made use of a naturally occurring ATF6α mutant that has been shown previously to result in a block in ER-to-Golgi trafficking (Chiang et al, 2017). Patients carrying a [D564G] mutation within the ATF6α lumenal domain suffer from a heritable blinding disease called achromatopsia. We created a cell line stably expressing ATF6α [D564G] and verified its lack of ER-to-Golgi trafficking following ER stress (Fig 7A). The Golgi form of ATF6α was observed for the wild-type protein but not the [D564G] mutant following DTT activation in the presence of the S1P inhibitor (Fig 7A, lanes 2 and 4). When the [D564G] mutant was stably expressed in the ERp18 KO cells, no ATF6-P cleavage product was observed in either the membrane or nuclear fraction (Fig 7B, lanes 3 and 7). To demonstrate that this mutant can be cleaved by S1P and S2P, we treated cells with brefeldin A, which results in the relocation of Golgi proteins to the ER (Niu et al, 2005). Under these conditions, cleavage of the [D564G] mutant of ATF6α in both wild-type and ERp18 KO cells was observed as evidenced by the appearance of cleavage products in the membrane and nuclear fractions (Fig 7B, lanes 2 and 4, and lanes 6 and 8). These results indicate that the ATF6-P cleavage product seen for wild-type ATF6α in the ERp18 KO cell line is generated following trafficking from the ER. In

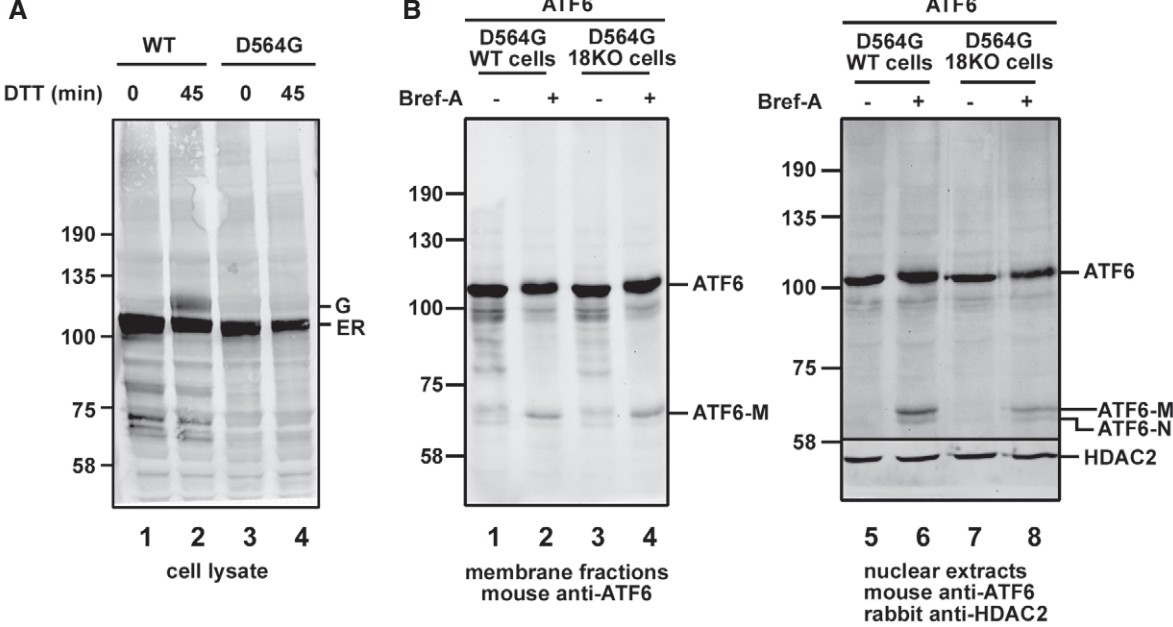

**Figure 7. ATF6α cleavage occurs in a post-ER compartment.**

A   HEK293T cells stably overexpressing wild-type ATF6α or a D564G mutant were treated with 10 mM DTT as indicated to induce ER stress in the presence of the S1P inhibitor. Cell lysates were prepared, separated by SDS–PAGE, and then probed with anti-ATF6α. Blot reveals the presence of O-linked glycan-modified Golgi-translocated ATF6 (designated G), which is absent for the D564G mutant, confirming a lack of ER-to-Golgi trafficking.

B   Wild-type and ERp18 KO cell lines stably expressing ATF6α D564G were either left untreated (−) or treated (+) with 5 µg/ml brefeldin A for 1 h prior to cell lysis. The anti-ATF6α Western blot of membrane fractions reveals proteolytic processing by S1P to produce ATF6-M, only in the brefeldin A-treated cells. Nuclear fractions confirm the presence of S2P cleavage product, ATF6-N, following brefeldin A treatment. Anti-HDAC2 was used as a loading control for the nuclear fraction.

Source data are available online for this figure.

addition, it demonstrates that S1P is fully functional in the ERp18 KO cell line.

### ERp18 KO leads to accelerated trafficking from the ER to the Golgi

The fact that ATF6α is cleaved in the Golgi in the absence of ER stress in the ERp18 KO cells would suggest that ERp18 has a role in retention. To address this point, we investigated whether there is any change to the kinetics of trafficking of ATF6α from the ER to the Golgi following ER stress. We followed trafficking after ER stress activation with DTT as this causes a rapid activation of ATF6α allowing the kinetics of transport to be evaluated. Trafficking was monitored by the appearance of Golgi-modified ATF6α in the presence of the S1P inhibitor (Fig 8). Note that the presence of the S1P inhibitor leads to the appearance of the ATF6-P cleavage product in the cell lines stably expressing ATF6α. In wild-type cells, the Golgi form of ATF6α first appeared at 30 min after DTT treatment with maximal Golgi form at 45 min (Fig 8A and G). In contrast, in the ERp18 KO cells most of the Golgi form was apparent after 30 min before declining at 45 and 60 min (Fig 8B and G). The appearance of the Golgi form of ATF6α was quantified to illustrate the kinetics of transport (Fig 8G). The absence of ERp18 led to an enhanced rate of trafficking, suggesting that ERp18 monitors the exit of ATF6α from the ER particularly after induction of ER stress. Expression of ERp18 in the KO cell line reversed the acceleration in ATF6α trafficking following DTT activation (Fig 8C and G). The DTT treatment may well affect the catalytic activity of ERp18 or the disulfide status of ATF6α; however, even in the presence of 10 mM DTT, ERp18 slows the exit of ATF6α from the ER. Taken together with the aberrant post-ER cleavage of ATF6α in the ERp18 KO cells, these results suggest that the role of ERp18 is to prevent the premature trafficking of forms of ATF6α that are poor substrates for S1P.

The role of ERp18 in retention of ATF6α could be related to the reduction of intra- or interchain disulfides within the lumenal domain. To evaluate this possibility, we determined the effect of mutating one or both of the cysteines within the ATF6α lumenal domain on the kinetics of transport (Fig 8D–G). For this experiment, we transiently expressed the ATF6 constructs in HEK293T cells, an approach which did not give rise to the ATF6-P product even in the presence of the S1P inhibitor. Note that the single cysteine mutants are still able to form interchain disulfides (Fig 4) whereas the double mutant cannot form any disulfides. For both the single mutants, the amount of ATF6α reaching the Golgi increased throughout the time course (Fig 8D and E). Very little of the double cysteine mutant was trafficked to the Golgi (Fig 8F), possibly due to a lack of correct folding. There was no evidence of a more rapid exit of ATF6α from the ER at early time points for any of the cysteine mutants as seen for the wild-type protein in the ERp18 KO cells. These results along with a lack of rapid trafficking in the presence of DTT indicate that the retention caused by the presence of ERp18 is not dependent on disulfide reduction.

## Discussion

The activation of ATF6α during the UPR is a highly regulated process requiring the release of ER retention and subsequent trafficking to

the Golgi apparatus. For trafficking to occur, proteins involved in retention dissociate, disulfides within the lumenal domain are reduced, ATF6α oligomers dissociate, and reduced monomeric ATF6α is packaged into COPII-coated vesicles for transport out of the ER (Fig 9; Nadanaka et al, 2007; Schindler & Schekman, 2009). The initial trigger for release comes from either unfolded proteins (Ye et al, 2000) or alterations to the lipid composition of the ER membrane (Tam et al, 2018). BiP is known to dissociate from ATF6α in the presence of unfolded proteins (Shen et al, 2002), which could potentially lead to the trafficking of disulfide-bonded ATF6α to the Golgi. In this work, we demonstrate a role for the oxidoreductase ERp18 in binding to ATF6α following ER stress. In ERp18 KO cells, ATF6α is trafficked more rapidly to the Golgi, suggesting it acts to retain ATF6α in the ER following BiP release. In addition, in the absence of ERp18 a fraction of ATF6α is cleaved by a post-ER-localized protease, distinct from S1P, resulting in a product (ATF-P) that accumulates and is not cleaved by S2P. To facilitate ER retention, ERp18 may bind to and mask previously identified Golgi-localization sequences within the ATF6α lumenal domain (Shen et al, 2002) to prevent trafficking and dissociate to allow loading into COPII vesicles. ERp18 is a disulfide exchange protein (Jeong et al, 2008) and can form mixed disulfides with ATF6α and so could directly reduce the disulfides within its lumenal domain thereby triggering its release. Hence, we have demonstrated that ERp18 performs an important regulatory role during the UPR, binding to ATF6α following BiP release thereby preventing premature exit from the ER.

How the association of BiP with ATF6α is regulated is unclear but is critical to activate the ATF6α signaling pathway. Previous work has shown that BiP forms a relatively stable association with the ATF6α lumenal domain and is unlikely to dissociate simply in the presence of unfolded proteins (Shen et al, 2005). In addition, an isolated ATF6α–BiP complex was resistant to dissociation in the presence of ATP, suggesting the requirement for additional cellular factors to facilitate release. Finally, using a semi-permeabilized cell assay that recapitulates vesicular budding and ATF6α exit from the ER, it was demonstrated that a reducing agent and ATP work synergistically to dissociate the complex (Schindler & Schekman, 2009). Collectively these studies indicate a requirement for additional factors to facilitate the release of BiP from ATF6α. BiP has been shown previously to be modified by the oxidation of thiols to either become sulfenylated or form a disulfide (Wei et al, 2012; Wang & Sevier, 2016). The requirement for a reducing agent for BiP dissociation from ATF6α might indicate the reduction of a thiol-modified BiP by a member of the PDI family. It is intriguing to consider that both PDIR (Higa et al, 2014) and ERp18 have been shown to be associated with ATF6α and required for its activation, strongly suggesting a role for disulfide exchange in regulating the stability of the BiP–ATF6α complex.

The reduction of disulfides within ATF6α is a prerequisite for trafficking from the ER (Nadanaka et al, 2007). We have demonstrated that this reduction can be carried out by some but not all members of the PDI family. Formation of mixed disulfides with substrate-trapping mutants of the PDI enzymes, including ERp18, indicates their ability to reduce ATF6α under non-stressed conditions, suggesting some redundancy. However, only ERp18 interacts and forms mixed disulfides with ATF6α under stress conditions, showing specificity in this stage of activation and trafficking. Hence, the initial reduction of the BiP–ATF6α complex could be facilitated

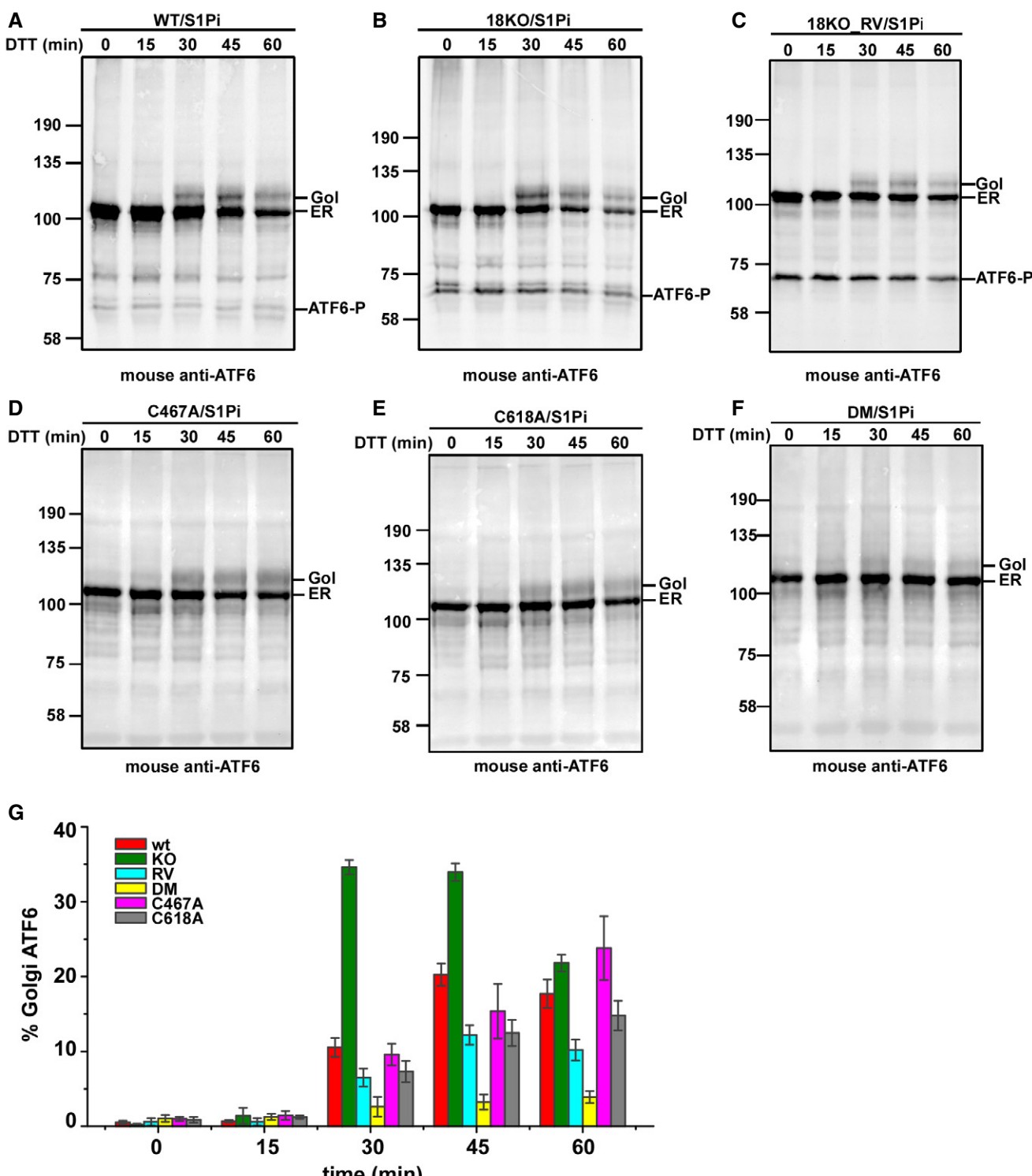

**Figure 8. Deletion of ERp18 promotes ER-to-Golgi trafficking of ATF6α.**

A–C Wild-type cells stably expressing ATF6α (A) or ERp18 KO cells stably expressing ATF6α (B), and ERp18 KO cells stably expressing ATF6α transiently transfected with ERp18 (RV cells) (C), were pre-treated with 30 μM PF429242 (S1P inhibitor) for 1 h prior to induction of ER stress with 10 mM DTT as indicated.

D–F Wild-type cells transiently expressing ATF6 C467A (D), ATF6 C618A (E), or ATF6 DM (F) were treated as above. Whole-cell lysates were separated by SDS–PAGE under reducing conditions and analyzed by Western blotting for ATF6α. The positions of unprocessed ER-localized ATF6α (ER) and the O-linked glycan-modified Golgi-translocated ATF6α (Gol) are indicated as is ATF6-P.

G The percentage of Golgi-ATF6α was quantified, and the error bars represent ± standard deviation for at least three independent experiments.

Source data are available online for this figure.

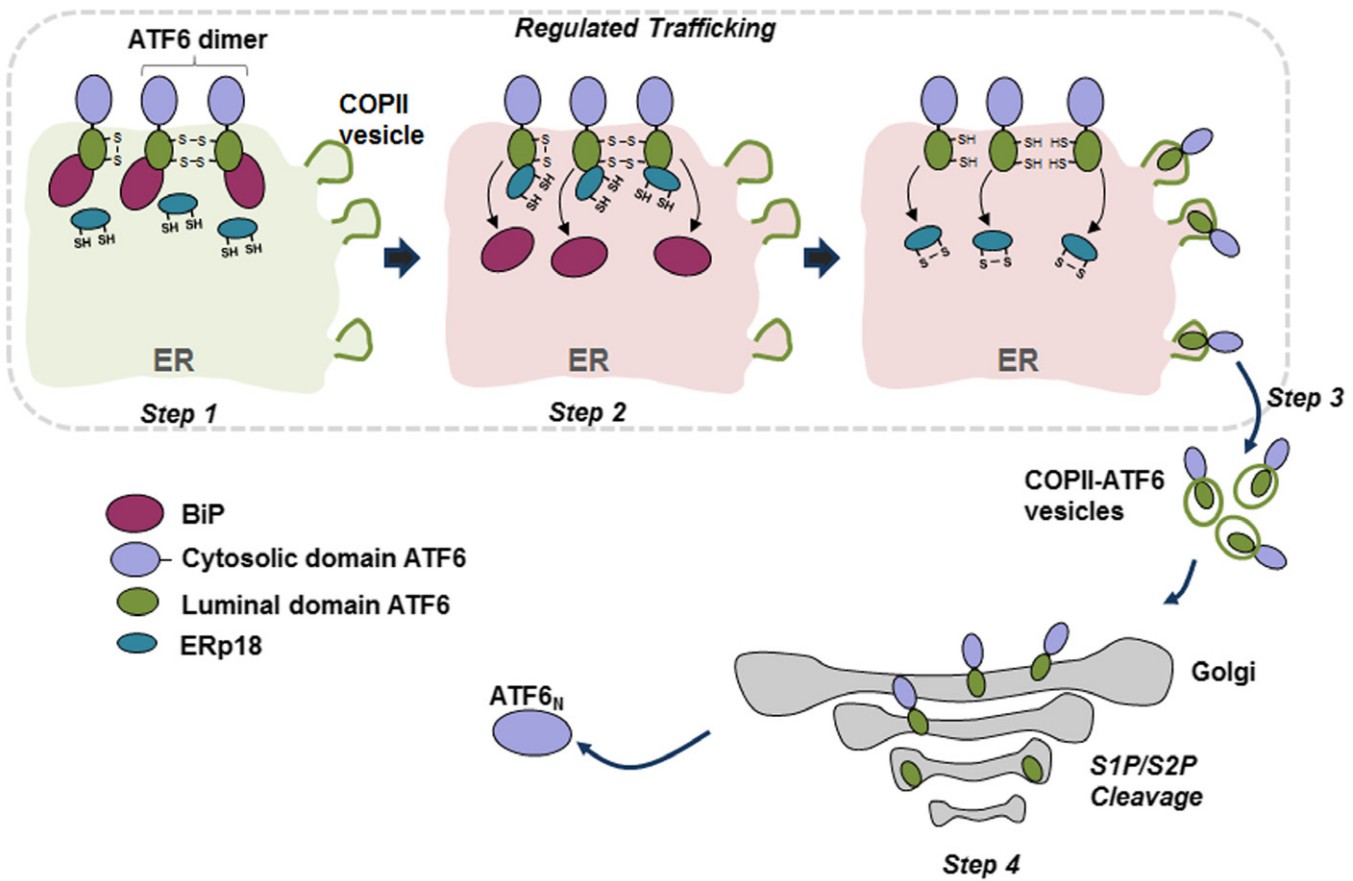

**Figure 9. ERp18 modulates trafficking and cleavage of ATF6α.**

Under no stress (light green shading), BiP binds to the lumenal domain of disulfide-bonded monomer and dimer ATF6α, preventing ER exit (Step 1). In Step 2, under conditions of ER stress (light red shading), ERp18 or other PDI family members could reduce ATF6α disulfides, resulting in the release of BiP from ATF6α. ERp18 binds, possibly before or after BiP release, and regulates ER exit. Reduced ATF6α is subsequently packaged into COPII vesicles and transported to the Golgi apparatus (Step 3). The regulated binding and release of chaperones and oxidoreductases ensures that only reduced ATF6α, which is the optimal substrate for S1P, is transported to the Golgi. Sequential cleavage by S1P and then S2P leads to membrane release and production of ATF6-N, respectively (Step 4). ATF6-N traffics to the nucleus to transactivate genes required to restore ER homeostasis. The fate of the lumenal domain is unknown but it maybe secreted. In the absence of ERp18, ATF6α rapidly exits the ER and is cleaved by an unknown post-ER-localized protease to produce a fragment that is inefficiently processed by either S1P or S2P.

by several members of the PDI family, but only ERp18 acts on ATF6α following release from BiP. The role of other proteins such as thrombospondin 4 in this overall process is unclear but could be to act as cargo receptors for packaging into COPII vesicles (Lynch *et al*, 2012).

The observation that ATF6α can be cleaved to ATF6-P in the presence of an S1P inhibitor has been reported previously, but the significance of this event has not been evaluated (Ye *et al*, 2000; Gallagher *et al*, 2016). It is known that the recognition of substrates for S1P is stringent in that they need to contain the consensus sequence for cleavage as well as the lumenal domain. Deletion of even a small portion of the lumenal domain results in a lack of cleavage (Chen *et al*, 2002). In addition, efficient cleavage by S1P requires ATF6α to be in a reduced state with no intra- or interchain disulfides (Nadanaka *et al*, 2007). Therefore, the formation of ATF6-P in the absence of ERp18 indicates a lack of recognition by S1P due to the premature trafficking of disulfide-bonded ATF6α. Previous studies indicate that ATF6-P should be a substrate for S2P as most of the lumenal domain

is required to prevent S1P-independent cleavage by S2P (Shen & Prywes, 2004). The persistence of this product and its lack of cleavage to ATF6-N by S2P would suggest that it is trafficked to a post-Golgi compartment making it inaccessible to S2P.

Studies into the activation of the other UPR sensors, Ire1 and PERK, have highlighted the complexity of the process, suggesting multiple layers of regulation including reversible post-translational modifications as well as co-factors regulating oligomerization, complex formation, and BiP binding (Carrara *et al*, 2015; Preissler *et al*, 2015; Amin-Wetzel *et al*, 2017; Hong *et al*, 2017; Sepulveda *et al*, 2018). It is of interest to note that the activation of each UPR sensor involves the dissociation of BiP; however, the consequence of BiP dissociation differs: Ire1 and PERK oligomerize to activate their kinase domains, whereas ATF6α dissociates to form monomers that can then exit the ER. Hence, for ATF6α there is a specific requirement for additional factors such as ERp18 to prevent ER exit following BiP release, thereby ensuring efficient processing to an active transcription factor.

# Materials and Methods

### Antibodies

The following primary rabbit polyclonal antibodies were used: anti-PDIR, anti-ERp57, anti-P5, and anti-GRP170 (all from ThermoFisher, cat. # PA3-007, PA3-009, PA3-008, and PA5-27655, respectively). In addition, a rabbit polyclonal antibody to the HA-tag (Sigma, cat. #H6908) and a goat polyclonal antibody to reticulocalbin (Santa Cruz, cat. #sc-109422) were used. Mouse monoclonal antibodies used were anti-ATF6 (Abcam, cat. #ab122897), anti-GAPDH (ThermoFisher, cat. #AM4300), anti-V5 (Invitrogen, cat. #R960-25), anti-myc clone 4A6 (Merck, cat. #05-724), anti-HA (Sigma, cat. #H3663), and anti-BiP (BD Biosciences, cat. #610979). A rabbit monoclonal anti-HDAC2 was also used (Abcam, cat. #32117). The following secondary antibodies were used for Western blotting: goat anti-mouse IRDye 800 (cat. #10751195), goat anti-mouse IRDye 680 (cat. #A32729), goat anti-rabbit IRDye 800 (cat. #13477187), and goat anti-rabbit IRDye 680 (cat. #35568, all from ThermoFisher). Secondary antibodies for immunofluorescence were sheep anti-rabbit-FITC (Sigma, cat. #F7512) and donkey anti-mouse-Texas Red (Abcam, cat. #ab6818).

### Cell lines

All HEK293T and HT1080 cells were maintained in DMEM supplemented with 10% FBS and 100 units/ml penicillin and 100 μg/ml streptomycin at 37°C in a 5% $CO_2$ incubator.

### DNA constructs

The pCGN-ATF6 plasmid containing an N-terminal HA epitope was obtained from Adam Benham (Durham University), originally a gift from Ron Prywes (Zhu *et al*, 1997). This construct was used as template to produce an ATF6α fragment containing an N-terminal HA-tag and a C-terminal V5-tag followed by a KDEL sequence and stop codon, using a two-step PCR with appropriate primers (Table EV1). This DNA fragment was then cloned into the NheI and NotI sites of pcDNA3.1. The single (ATF6α C467A and ATF6α C618A) and double (C467A/C618A, ATF6α DM) cysteine-to-alanine mutations in the lumenal domain of ATF6 were generated using the QuikChange site-directed mutagenesis kit (Agilent) with the appropriate primer pairs (Table EV1). The ATF6α S1P mutant R415A/R416A was synthesized by GenScript. The human PDI CXXA trap mutants have been described previously (Jessop *et al*, 2009; Oka *et al*, 2013). Briefly, the coding sequences including a C-terminal V5-epitope tag followed by a KDEL sequence and a stop codon were amplified by PCR and then cloned into pcDNA3.1. The human TMX1 CXXA trap mutant containing cysteine-to-alanine mutation in the active site and a C-terminal HA-tag, cloned into pcDNA3.1(+) vector (Pisoni *et al*, 2015), was a gift from Maurizio Molinari (IRB, Bellinzona, Switzerland).

The wild-type EGFP-tagged human ATF6α S1P (R415A/R416A) was obtained from Addgene (Chen *et al*, 2002). The ATF6α mutant was cloned into pEGFP-C3 vector to allow expression of a fusion protein containing an N-terminal EGFP protein. The EGFP-ATF6-S1P C618A mutation was generated using the QuikChange site-directed mutagenesis kit (Agilent) with the appropriate primer pairs

(Table EV1). The wild-type human reticulocalbin 1 (RCN1) cloned into the pCAGGS vector was a kind gift from Masayuki Ozawa, Kagoshima University, Japan (Ozawa and Muramatsu, 1993).

### Cell culture and transfections

HEK293T cells were transfected with plasmid DNA at 80–90% confluence using either MegaTran (OriGene) or Lipofectamine 2000 (ThermoFisher Scientific) transfection reagent. To generate stable cell lines, the transfected cells were placed on antibiotic selection for approximately 2–4 weeks until colonies appeared. Positive clones were identified by Western blotting.

### Cell lysis, immunoisolation, and Western blots

Cells were harvested by treatment with trypsin–EDTA (Gibco) or by scrapping using a rubber policeman (Greiner Bio-One). They were collected by centrifugation at $1,000 \times g$ for 5 min and then washed twice by ice-cold PBS. The cells were then resuspended in lysis buffer [1% (v/v) Triton X-100, 50 mM Tris–HCl (pH 7.4), 150 mM NaCl, 2 mM ethylenediaminetetraacetic acid (EDTA), and 0.5 mM phenylmethylsulfonyl fluoride (PMSF)], incubated on ice for 10 min, followed by centrifugation at $16,000 \times g$ to obtain the post-nuclear supernatant.

Prior to immunoisolation, the post-nuclear supernatant was precleared by incubating with protein A Sepharose beads (Generon) for 30 min at 4°C. The mixture was then precleared by centrifugation at $14,000 \times g$ for 1 min and the supernatant incubated with protein A Sepharose beads and the appropriate antibody or GFP-Trap (Chromotek, cat. #ABIN509407) for 16 h at 4°C. Immunoisolated material was washed three times in lysis buffer or in SDS wash buffer (lysis buffer supplemented with 350 mM NaCl and 0.5% SDS) for the GFP-Trap isolations. Samples were boiled at 95°C for 5 min in SDS–PAGE sample buffer [200 mM Tris–Cl (pH 6.8), 3% SDS, 10% glycerol, 1 mM EDTA, and 0.004% bromphenol blue] prior to SDS–PAGE under either reducing (treated with 50 mM DTT) or non-reducing conditions.

For Western blotting, proteins were transferred to nitrocellulose membrane (Li-Cor Biosciences), which were blocked in 5% (w/v) non-fat dried skimmed milk in TBST [Tris-buffered saline with Tween-20: 10 mM Tris, 150 mM NaCl (pH 7.5), and 0.1% (v/v) Tween-20] for 60 min. Primary antibodies were diluted in TBST, and incubations were carried out for 16 h at either 4°C or room temperature. IRDye fluorescent secondary antibodies were used for detection, typically at 1:5,000 dilutions. Blots were scanned using an Odyssey SA imaging system (Li-Cor Biosciences).

### Mass spectrometry

Confluent untransfected HEK293T and HEK293T cells stably expressing HA-ATF6-V5 were either left untreated or treated with 5 μg thapsigargin for 60 min. The cells were then treated with 2 mM dithiobis(succinimidyl propionate; DSP) and incubated at room temperature for 30 min to form protein cross-links followed by 20 mM Tris (pH 7.5) for 15 min at room temperature to quench the reaction. The cells were collected by centrifugation at $1,000 \times g$ for 5 min and then rinsed twice with ice-cold PBS supplemented with 20 mM NEM. Post-nuclear supernatant was prepared in lysis

buffer containing 1% (v/v) Triton X-100, 50 mM Tris–HCl (pH 7.4), 150 mM NaCl, 2 mM ethylenediaminetetraacetic acid (EDTA), and 0.5 mM phenylmethylsulfonyl fluoride (PMSF) supplemented with EDTA-free protease inhibitor tablet.

Cell lysates were precleared by incubation with protein A Sepharose (PAS) for 30 min at 4°C, before incubation with anti-V5-conjugated agarose beads (Sigma) for 16 h at 4°C. The beads were washed three times with lysis buffer supplemented with 0.5% SDS and then incubated with 10 mM DTT (prepared in 25 mM ammonium bicarbonate) for 10 min to elute cross-linked complexes. Trypsin (0.3 µl, 0.2 ng/µl, Promega, sequencing grade) was added to the protein mixture and the solution incubated at 37°C overnight, to allow complete digestion. A portion of the resultant peptides were then injected on an Acclaim PepMap 100 C18 trap and an Acclaim PepMap RSLC C18 column (Thermo-Fisher Scientific), using a NanoLC Ultra 2D Plus loading pump and a NanoLC AS-2 autosampler (Eksigent). The peptides were held on the trap and washed for 20 min and were eluted with a gradient of increasing acetonitrile, containing 0.1% formic acid (2–20% acetonitrile in 90 min, 20–40% in a further 30 min, followed by 98% acetonitrile to clean the column, before re-equilibration to 2% acetonitrile). The eluate was sprayed into a TripleTOF 5600 + electrospray tandem mass spectrometer (AB Sciex, Foster City, CA) and analyzed in Information Dependent Acquisition (IDA) mode, performing 250 ms of MS followed by 100 ms of MS/MS analyses on the 20 most intense peaks seen by MS. The MS/MS data file generated via the "Create mgf file" script in PeakView (Sciex) was analyzed using the Mascot search algorithm (Matrix Science), against the NCBInr database (August 2016) considering both all species (93482448 sequences) and restricting the search to Homo sapiens (331464 sequences), trypsin as the cleavage enzyme and N-ethylmaleimide, hydrolyzed N-ethylmaleimide modifications of cysteine, thioacyl modification of lysines and N-termini, and methionine oxidation all as variable modifications. The peptide mass tolerance was set to 20 ppm and the MS/MS mass tolerance to ±0.05 Da. A protein was accepted as identified if it had 2 or more peptides with Mascot Ion Scores above the identity threshold ($P < 0.05$), and for those proteins identified by only two peptides, the MS/MS spectral assignments match most of the peaks in the MS/MS spectra.

For MS analyses of gel bands, the gel band was excised and cut into 1-mm cubes. These were then subjected to in-gel digestion, using a ProGest Investigator in-gel digestion robot (Genomic Solutions, Ann Arbor, MI) using standard protocols (Shevchenko *et al*, 1996). Briefly, the gel cubes were destained by washing with a 1:1 mixture of 30 mM potassium ferricyanide and 100 mM sodium thiosulfate and subjected to reduction and alkylation before digestion with trypsin at 37°C. The peptides were extracted with 10% formic acid and concentrated down to 20 µl using a SpeedVac (Thermo Savant). A portion of the resultant peptides were then injected on an Acclaim PepMap 100 C18 trap and an Acclaim PepMap RSLC C18 column (ThermoFisher Scientific), using a NanoLC Ultra 2D Plus loading pump and a NanoLC AS-2 autosampler (Eksigent). The peptides were eluted with a gradient of increasing acetonitrile, containing 0.1% formic acid (5–40% acetonitrile in 5 min, 40–95% in a further 1 min, followed by 95% acetonitrile to clean the column, before re-equilibration to 5% acetonitrile). The eluate was sprayed into a TripleTOF 5600 + electrospray tandem mass spectrometer (AB Sciex) and analyzed in Information Dependent Acquisition (IDA) mode, performing 250 ms of MS followed by 100 ms of MS/MS analyses on the 20 most intense peaks seen by MS. The MS/MS data file generated via the "Create mgf file" script in PeakView (Sciex) was analyzed using the Mascot search algorithm (Matrix Science), against the NCBInr database (August 2016) considering both all species (93482448 sequences) and restricting the search to Homo sapiens (331464 sequences), trypsin as the cleavage enzyme and N-ethylmaleimide, hydrolyzed N-ethylmaleimide modifications of cysteine, thioacyl modification of lysines and N-termini, and methionine oxidation all as variable modifications. The peptide mass tolerance was set to 20 ppm and the MS/MS mass tolerance to ±0.05 Da.

### Analyses of ATF6–PDI mixed-disulfide complexes

HEK293T cells were co-transfected with ATF6α or its cysteine mutants and PDI CXXA trap mutants. After 24 h following transfection, cells were either left untreated or treated with either 5 µM thapsigargin, 5 µg/ml tunicamycin, or 20 µM MG132 for 1 h prior to cell lysis, immunoisolation, and Western blotting.

### Silver staining

HEK293T cells were transiently transfected with either EGFP-ATF6 or EGFP-ATF6 S1P only or co-transfected with ERp18 substrate-trapping mutant. Whole-cell lysates were prepared 24 h after transfection and GFP-ATF6 immunoisolated as described above. Immunoisolated material was separated by SDS–PAGE under non-reducing conditions and bands detected by silver staining (Gharahdaghi *et al*, 1999).

### CRISPR/Cas9-based knockout of ERp18

The single guide RNA (gRNA) matching the genomic targets was designed using the CRISPR Design Tool—nickase analysis (crispr.mit.edu). The CRISPR/Cas9 expression vectors were prepared as reported previously (Kabadi *et al*, 2014). Briefly, single gRNAs were engineered with compatible overhangs (Table EV1), annealed, and phosphorylated before cloning into BbsI sites of compatible expression vectors. sgRNA_14 was cloned into the mU6 vector, while sgRNA_8 and sgRNA_10 were cloned into the hU6 vector. The integrity of the constructs was confirmed by DNA sequencing. Two sgRNAs, selected based on the nickase analysis, were combined, as indicated below, and then cloned into the pMulti-Cas9D10A-GFP vector using Golden Gate Assembly.

1  sgRNA_14 and sgRNA_8
2  sgRNA_14 and sgRNA_10

To generate an ERp18 knockout, 5.5 µg of the individual pMulti-Cas9D10A-GFP vectors was co-transfected with 0.5 µg pPur (puromycin-resistant vector; Clontech) into HEK293T cells stably expressing ATF6. After 24-h incubation in media, cells were transferred to media supplemented with 1 µg/ml puromycin and incubated for a further for 5 days, then subsequently in media without puromycin until colonies appeared (about 12 days later). Positive ERp18 knockout cells were identified by Western blotting using the mouse anti-ERp18.

### RNA extraction and qPCR

Total RNA was extracted from confluent cells from 6-cm dishes either untreated or treated with 1 μM thapsigargin for 16 h using the RNeasy Mini Kit (Qiagen) according to the manufacturer's instructions. cDNA synthesis was carried out using the SuperScript II Reverse Transcriptase Kit (Invitrogen) according to the manufacturer's protocols. Quantitative real-time PCR was carried out with 500 ng cDNA using the Brilliant III Ultra-Fast SYBR Green qRT-PCR Master Mix (Agilent Technologies) in a StepOnePlus system (Applied Biosystems). The primer pairs used are detailed in Table EV1.

### Analyses of ATF6α transcriptional targets in ERp18 knockout cells

Wild-type and ERp18 KO HEK293T cell lines stably overexpressing recombinant ATF6 wild type were used. To generate the ERp18 KO reversion (18KO RV) cell line, the ERp18 KO HEK/ATF6 cells were transiently transfected with a pcDNA3/ERp18 plasmid (Jessop *et al*, 2009). Confluent cells were either left untreated or treated with 1 μM thapsigargin for 16 h. Whole-cell lysates were separated by SDS–PAGE under reducing conditions followed by Western blotting. Quantification of band intensities was carried out with the Li-Cor Image Studio version 4 software and histograms prepared using Origin 6.0 professional software.

### ER stress-induced cleavage of ATF6α

Wild-type or ERp18 knockout HEK293T cells overexpressing ATF6α were either left untreated (−) or treated (+) for 1 h with 30 μM PF429242 (S1P inhibitor) before addition of 10 mM DTT or 5 μM thapsigargin to induce ER stress. For the brefeldin A (5 μg/ml) treatment, cells were incubated with the drug for 1 h prior to cell lysis. Post-treatment, cells were harvested and subjected to differential centrifugation as described previously (Gallagher *et al*, 2016). Briefly, cells were washed once with 3 ml PBS supplemented with EDTA-free protease inhibitor tablet. Cells were then scraped off the dish in 2.9 ml ice-cold PBS, using a rubber policeman (cell scraper) into a 15-ml tube. Samples were centrifuged at $3,000 \times g$ for 5 min, and supernatant was removed. Cells were resuspended in 1 ml ice-cold buffer A [10 mM HEPES (pH 7.4), 250 mM sucrose, 10 mM KCl, 1.5 mM $MgCl_2$, 1 mM EDTA, and 1 mM EGTA] supplemented with protease inhibitor, transferred to 1.5-ml tube, and left on ice for 10 min. Cells were lysed by passing through a 23-gauge needle attached to a 1-ml syringe, 20 times, followed by centrifugation at $1,000 \times g$ for 7 min at 4°C to obtain the nuclear pellet; the supernatant contains the cytosolic and membrane components. The pellet was washed once in 500 μl buffer A (plus protease inhibitor) and then centrifuged at $1,000 \times g$ for 7 min at 4°C. The pellet was then resuspended in 100 μl buffer B [10 mM HEPES (pH 7.6), 2.5% glycerol, 420 mM NaCl, 1.5 mM $MgCl_2$, 1 mM EDTA, and 1 mM EGTA] supplemented with protease inhibitor, followed by incubation for 60 min at 4°C. The samples were centrifuged at $100,000 \times g$ for 30 min at 4°C to obtain a supernatant containing the nuclear extract.

To obtain the membrane components, the supernatant from the initial $1,000 \times g$ centrifugation was centrifuged at $100,000 \times g$ for 30 min at 4°C. The pellet obtained was resuspended in 100 μl buffer B (+protease inhibitor). The nuclear extract and membrane fractions were separated by SDS–PAGE under reducing conditions and ATF6 identified by Western blotting.

### ER-to-Golgi trafficking of ATF6α

Wild-type, ERp18 KO, and 18KO RV HEK293T/ATF6 wild-type cells were used to follow ER-to-Golgi trafficking. The individual ATF6α cysteine mutants were transiently transfected into HEK293T cells. Confluent cells were pre-treated with 30 μM PF429242 (S1P inhibitor) for 60 min prior to inducing ER stress with 10 mM DTT. Whole-cell lysates were separated by SDS–PAGE under reducing conditions and ATF6 identified by Western blotting as described above. Quantification of band intensities was carried out with the Li-Cor Image Studio version 4 software and histograms prepared using Origin 6.0 professional software.

### Metabolic labeling and pulse-chase analysis

Wild-type and ERp18 KO HEK293T cells stably overexpressing ATF6 were incubated in medium lacking methionine and cysteine for 30 min and pulse-labeled for 30 min with 22 μCi/ml of EXPRESS35S Protein Labeling Mix (PerkinElmer). The cells were rinsed twice in PBS to remove the radiolabel and then incubated in complete medium to initiate the chase periods. Subsequently, cells were washed twice with PBS supplemented with 20 mM NEM and lysed in cell lysis buffer containing 20 mM NEM. Samples were subject to centrifugation at $16,000 \times g$ at 4°C to obtain the post-nuclear supernatant. ATF6α immunoisolation was carried out as described above using anti-ATF6 (Abcam). Samples were separated under non-reducing and reducing SDS–PAGE gels, fixed, dried, and exposed to a phosphorimager plate for 72 h. Radioactivity was detected using a Fujifilm FLA-7000 Phosphorimager.

**Expanded View** for this article is available online.

### Acknowledgements

This work was funded by the Wellcome Trust grant numbers 103720 (NJB) and 105614 (OBVO). We thank Adam Benham (Durham University), Maurizio Molinari (IRB Bellinzona, Switzerland), Adam West, Pablo Cabrero, Anthony Dornan, and members of the Dow/Davies Laboratory (University of Glasgow) for advice and reagents, and the mass spectrometry facilities at Glasgow and St Andrews universities.

### Author contributions

OBVO and NJB designed the experiments and wrote the paper. OBVO performed the experiments with assistance from ML, JR, WT, and M-AP.

### Conflict of interest

The authors declare that they have no conflict of interest.

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
