## [Review Process File · The EMBO Journal]

ERp18 regulates activation of ATF6 α during unfolded protein response

Ojore B. V. Oka, Marcel van Lith, Jana Rudolf, Wanida Tungum, Marie-Anne Pringle, and Neil J. Bulleid.

Review timeline:

Submission date:	23 rd October 2018
Editorial Decision:	23 rd November 2018
Revision received:	2 nd April 2019
Editorial Decision:	24 th April 2019
Revision received:	3 rd May 2019
Accepted:	20 th May 2019

Editor: Anne Nielsen / Elisabetta Argenzio

Transaction Report:

1st Editorial Decision

23rd November 2018

Thank you for submitting your manuscript for consideration by the EMBO Journal. It has now been seen by three referees whose comments are shown below.

As you will see from the reports, the referees all express interest in the findings reported in your manuscript although they also raise a number of concerns that you will have to address - either experimentally or with text changes - before they can support publication of the manuscript here.

For the revised manuscript I would particularly ask you to focus your efforts on the following points:

-> Please follow ref #1's suggestion to rule out ATF6a misfolding in the ER. In addition, this referee points to a number of cases where clarifications on the figures, the underlying experiments or the interpretations are needed. I would ask you to clarify all these points.

-> Ref #2 mainly asks for text clarifications that should be straightforward to include.

-> For ref #3, I find the comments constructive overall but I want to mention that the inclusion of genome-wide RNAseq data (point #1) is not a requirement from our side. For point #3, I would suggest that you include new data on this point if you have it readily available, otherwise I'd be happy to discuss what you could include/discuss on the interplay between ERp18, ATF6 and BiP.

Given the referees' overall positive recommendations, I would like to invite you to submit a revised version of the manuscript, addressing the comments of all three reviewers.

REFeree REPORTS

Referee #1:

In their study entitled ERp18 Regulates the Activation of ATF6a during the Unfolded Protein Response Oka et al. identify ERp18 as a new regulator of ATF6a trafficking and processing. There is a real gap of knowledge about how ATF6a is activated in the ER, inducing its trafficking. Thus, the study is of high and general interest. However, at several places, the method/data presentation/controls are too superficial and conclusions not justified, which dampened the enthusiasm of this reviewer. Thus, several important points need to be addressed to potentially further proceed with the manuscript:

Major points:

- In Figure 1A, constructs and schematics deserve more explanation and C618A is mis-indicated; 1B: to make the point of ER localization, co-localization with an ER marker needs to be shown; and why is ATF6a called "recombinant" here?
- The normal behavior of the tagged ATF6a construct (trafficking, cleavage) needs to be controlled for. In particular, since the mass spec experiments identified a lot of ER chaperones, suggesting substrate-like interaction with the ATF6a construct and some known interactors were missing. Along these lines: the authors should include experiments to show ERp18 is not simply a folding PDI/chaperone for ATF6a
- Why is no ERp18 data shown in Figure 1? 1D could e.g. be replaced by this data.
- Figure 2A indeed shows multiple bands upon co-expression of substrate-trap PDI family members; but their molecular weights, number and heterogeneities warrant further controls, e.g. ATF6a IPs and blots against the PDI family members. And why are the gels scaled differently, but O/D run at the same height? Furthermore, the data in 2B is not convincing as presented. The exposure of the blot is too different from 2A, for P5 bands may in fact be there.
- In Figure 3, why do the authors now switch from anti-V5 to anti-HA?
- The authors claim that "The only protein which exclusively associated with ATF6a after ER stress was the PDI family member ERp18". In Figure 3A, however interaction between ATF6a and ERp18 is shown under any condition, and may even be reduced under stress conditions. And similar to the comments above: the various molecular weights, number and heterogeneities of species warrant further controls. In C, the lower ATF6 band also deserves a double star, as it will most likely be retrotranslocated and deglycosylated as the authors write in the main text.
- The data presented in Figure 3 do not make a very compelling case for a special role of ERp18: its binding is also reduced under ER stress, similar to e.g. ERp57 $-/+$ Tm; along the same lines: why are only Tg data shown in D, and what discriminates the bands with an asterisk for PDIR from the ones not given an asterisk?
- The authors claim that data presented in Figure 3 shows that "ERp18 remains catalytically active towards ATF6a even after ER stress"; how is this discriminated from lack of release?
- Figure 4 presents some very interesting findings. But how do the authors come to the conclusion that ATF6a oligomers do not exist simply based the running behavior of ERp18 adducts/species of the disulfide mutants? Other interpretations are also possible.
- In Figure 5B, controls of basal levels, w/o Tg, would also be needed.
- For many experiments, the authors use DTT to induce ER stress/ATF6 trafficking. If ERp18 & ATF6 form a disulfide-bonded complex, DTT will likely break this. The authors should comment on this.
- The model in Figure 9 is not entirely clear: what happens to the luminal domain of ATF6 in this model? And where do the authors conclude from, that BiP needs to be released first before ERp18 acts on ATF6?
- To further verify the authors' model: Does the Cys-double mutant in ATF6 lose completely binding to ERp18, and dependency on ERp18 for trafficking (in ERp18 k/o cells)?

Minor points:

- In the introduction, the authors call the arms of the UPR the "integrated stress response"; this term is normally used for different pathways converging on eIF2a and thus should better not be used in this context. Furthermore, what do the authors mean by "the translational programme activated by PERK"? Also, a main function of ATF6, XBP1 splicing, is missing. The authors thus should rework the first paragraph of their introduction.
- Figure 5G is not mentioned in the legend and text; also, the effect of expressing ERp18 would be nice to show here.

- In Figure 6A, the HDAC marker should also be shown.
- In Figure 7, it is not clear where SIP inhibitors were used and where not.

Referee #2:

This manuscript describes the finding that ERp18, a small PDI family member about which little is known functionally, helps regulate the transport of ATF6 α to the Golgi and thus affects the induction of the ATF6 α branch of the unfolded protein response. This is an interesting and topical observation. Overall the study appears to have been well-executed and the data support the conclusions. The observation that ATF6 α , when mis-regulated, can arrive at the Golgi in a form that is then unable to be processed properly is intriguing.

Nevertheless, there are a few points where the manuscript can be improved, as listed below.

On page 6, the following statement is confusing. "The interactions were specific to ATF6 α as a signal was either absent in the untransfected cell line (BiP, reticulocalbin and P5) or greatly enhanced in the transfected cell-line (Grp170)." One could say instead, "The interactions were specific to ATF6 α , as the signals were much lower or undetectable in the untransfected cell line."

On page 8 and in Figure 2A, it would be helpful if the M, D, and O abbreviations could be given in the figure legend rather than the main text. (They appear in the legends to Figures 3 and 4.)

Is the following statement on page 9 solid? "Single mixed disulfides species were seen with either cysteine mutant in the absence of ER stress (Fig. 4A, lanes 3 and 5) indicating that ERp18 reduces interchain disulfides formed via either C467 or C618." If this reviewer understands the logic correctly, the authors assume that the ERp18 trapping mutant must be attacking a disulfide to form the mixed disulfide with ATF6 α . Sounds reasonable. They also seem to assume that, since there are only two cysteines in the ATF6 α luminal domain and since the mixed disulfide with ERp18 forms even when one of these cysteines is absent, the remaining ATF6 α cysteine must form a bond with a copy of itself on another ATF6 α molecule, because there is no other option. Is that a fair conclusion, though? Could the remaining cysteine form a mixed disulfide with glutathione or something else in the single-cysteine mutants, and this mixed disulfide is then attacked by ERp18? Or did the reviewer miss the point?

Actually, the paragraph beginning "The mobility of..." continues to be confusing. Part of the problem is poorly formed sentences (such as too many "withs" in the first one). Also, the authors state that they are looking for additional proteins, but they don't fully close the door they opened with this question. Rather than state that "only ATF6 α was present..." they could say instead that no other proteins were detected, and that they therefore favor the alternative explanation for the mobility differences. Another source of confusion is that the text questions the identities of the dimer and oligomer for the wild-type protein, but Fig. 4C, which is cited right afterwards, does not show the new interpretation of the dimer/oligomer, only the mixed disulfides with ERp18.

On page 10 the authors write, "The induction of BiP mRNA was dramatically attenuated in the KO cells whereas other UPR target gene induction was unaffected (Fig. 5B)." Then on the following page they refer to the attenuation of induction of ERp72. Is ERp72 not attenuated on the transcriptional level, or was it not tested? In any case, the way these two bits of information are presented appears contradictory.

Should "emPAI" be defined?

In Figure 1A, there is a problem with C618A. Also, the cysteine-to-alanine mutations are listed only as A467 and A618 under the primary structure maps, whereas the arginine-to-alanine mutants are listed as R415A and R416A.

In Figure 1C, the issue with RCN1 is not clear. Why was it transiently transfected?

"ATF6" appears twice, unnecessarily, in the last sentence of the Figure 7A legend.

Stylistic point: Commas are missing/misplaced/misused in many cases. For one example of many, the Figure 9 legend reads, "ATF6, traffics to the nucleus..."

Referee #3:

Activating transcription factor 6 (ATF6) is an unfolded protein response (UPR) transcription factor with a single-span endoplasmic reticulum (ER) transmembrane domain. In response to an accumulation of unfolded proteins in the ER, two major molecular events trigger the activation of ATF6: (1) the disruption of inter- and intra-disulfide bonds formed between two cysteines within the ER luminal portion of ATF6 and (2) the dissociation of BiP, a major ER chaperone, from the ER luminal domain of ATF6. Although previous work established the presence of these events, we still lack a detailed understanding. For example, we still do not know the identities of the ER oxidoreductases that reduce the ATF6 disulfide bonds. Furthermore, the temporal relationships between BiP dissociation and reduction of disulfide bonds remain elusive.

In this study, Dr. Neil Bulleid and his colleagues identified and subsequently confirmed several ER-resident oxidoreductases associated with ATF6 using mass spectrometry (mass spec) of immunoprecipitated fractions with an anti-ATF6 antibody. The authors used elegant substrate trapping experiments with a mutagenized form of oxidoreductases identified by mass spec, together with one or two cysteine mutants of ATF6, to find that ERp18 remains associated with the ATF6 ER luminal domain even after ER stress induction. Furthermore, this interaction with ERp18 occurred specifically via the ATF6-Cys467 residue. This was a rather unexpected result, as oxidoreductases should dissociate from ATF6 after reduction of disulfide bonds within the ATF6 ER luminal domain. Thus, these results suggest unique functions of ERp18 in the activation of ATF6 beyond disruption of ATF6 disulfide bonds.

In subsequent experiments in ERp18 knockout (KO) cells, the authors uncovered that ATF6 could be cleaved by a protease(s) beyond S1P/S2P, producing an aberrant ATF6 fragment (ATF6-P) in the Golgi. The authors also found that COPII trafficking of ATF6 from the ER to the Golgi was fastened in the absence of ERp18, leading to aberrant ATF6 cleavage that cannot be further cleaved by S2P to release the ATF6 transcription activation domain from the Golgi membrane. Consequently, ERp18 KO cells were unable to induce the transcription of GRP78, one of the ATF6 transcription target genes. Taken together, the authors concluded that ERp18 plays an important role in ensuring the appropriate transport of ATF6 from the ER to the Golgi.

Overall, the experiments were both well designed and well executed to support the authors' conclusions. The findings are important and will be of interest for general readers of EMBO Journal, unveiling a previously unrecognized regulatory step of early ATF6 activation. The following are specific comments.

#1. In Figure 5, using ERp18 KO cells, the authors showed that GRP78 transcript levels did not increase upon ER stress. Additionally, ERp72 protein levels remained unchanged in ER-stressed ERp18 KO cells. These results suggest that ERp18-mediated regulation of ATF6 is important. However, to strengthen this conclusion, the authors should perform the following additional experiments: (i) As ATF6 is a transcription factor, the authors should examine mRNA levels of ERp72, instead of protein levels; (ii) More importantly, the authors should perform genome-wide transcription analysis of ER-stressed or unstressed ERp18 KO cells. These additional experiments will extend the scope of the impact of ERp18 KO on ATF6-specific transcription in general, and not limited to GRP78 and ERp72.

#2. Some of the ATF6 species shown on SDS-PAGE are very close in size (e.g., ATF6-P vs. ATF6-M in Figure 6D and ATF6-P vs. ATF6-N in Figure 6E). How was the identity of each band confirmed? In addition, in Figure 6A lanes 2 and 3, several bands are in the molecular weight range of 58 and 75 kDa. How was the identity of ATF6-P determined?

Furthermore, the authors should provide an explanation as to why ATF6-FL or ATF6-P should be

present in a nuclear fraction. In the nuclear fraction of ERp18 KO cells, why did ATF6-FL disappear?

#3. What is the potential role of ERp18 in regulating ATF6 transport from the ER to the Golgi? Does ERp18 ensure that BiP-dissociated ATF6 monomers are identified? Can BiP-associated ATF6 enter into COPII vesicles in the absence of ERp18? Further, does ERp18 enter into COPII vesicles with ATF6? Or alternatively, does ERp18 dissociate from ATF6 prior to being packaged into COPII vesicles?

In ERp18 KO cells, why was ATF6-P generated constitutively even in the absence of ER stress (Figure 6B)?

#4. The authors should examine whether the effect of ERp18 is specific to ATF6. Is transport or activation of SREBP impacted in ERp18 KO cells in response to low cholesterol levels?

#5. What is the relationship between S1P and the alternative protease? Specifically, this is puzzling in Figure 6E lane 3 where ATF6-N is produced upon DTT treatment if S1P (and S2P) is functional but inhibition of S1P causes ATF6-P to remain in the membrane fractions. In Figures 6A and B, significant levels of ATF6-P are present in both membrane and nuclear fractions. In contrast, ATF6-P in WT cells seems to exist only in the membrane fractions.

#6. In Figure 8C, although transfection of ERp18 into ERp18 KO cells recovered the kinetic appearance of ATF6-Golgi, ATF6-P is still generated. This contrasts with WT cells, in which ATF6-P was produced at minimum levels (Figure 8A). What was the expression level of transfected ERp18? If potential differences come from different expression levels of ERp18, why should ATF6 transport be recovered without affecting the generation of ATF6-P?

#7. In Figure 7, the authors conclude that S1P is active in ERp18 KO cells by re-localizing S1P to the ER via bref A treatment. However, when transport of ATF6 to the Golgi is blocked in the D564G ATF6 mutant, ER-localized S1P functions more effectively with WT than it does for D564G ATF6. What is a possible explanation for this?

Minor points:

#1. There is no mention of Figure 5G in the results section. The authors should include this figure panel in the results section.

#2. Page 23, under methods section, 'CRISPR/Cas9-based knockout of ERp18', shouldn't the last sentence, "the monoclonal anti-ERp19 (Abcam)", be anti-ERp18 (Abcam)?

1st Revision - authors' response

2nd April 2019

Point by point response to reviewers:

Reviewer #1:

We note the reviewer's generally positive comments and thank her/him for the suggested modifications, which we have attempted to address by further experimentation and clarification. Our responses are detailed below.

Major points:

- We have altered Fig. 1A to correct mistakes and provided more details in the figure legend. We have replaced Fig. 1B with a panel showing co-localisation of ATF6 with an ER marker. We have also adjusted the figure legend to remove the reference to recombinant ATF6.
- To investigate whether ERp18 is involved in the folding of ATF6 α we carried out a pulse chase experiment to see if there was any delay in the acquisition of disulfides in the ERp18 KO cell-line. We see no difference in the formation of

disulfides between the wild type and ERp18 KO cells. We include this figure in the expanded view (Figure EV1) and modified the results section accordingly (page 11).

- We have now included an additional figure in 1D which shows the association of ERp18 with ATF6 α in unstressed cells which increases after ER stress.
- The controls requested by the reviewer (ATF6 α IPs and western blots to PDI family members) are included in Fig. 3D. Fig. 2 clearly shows additional ATF6 α species that are present in the transfected but not untransfected cells. We have repeated the gels for Figure 2A using the HA epitope rather than V5 for consistency with Fig 3. In addition, as these are on the same gel there is no heterogeneity in migration distance (as was the case when they are run on separate gels for differing lengths of time). Fig. 2B has been repeated, with lower loading so that we can expose the blots for longer to increase the background as requested.
- We have repeated Figure 2 with the HA antibody for consistency with Figure 3. Note that the experiment in Figure 2 was carried out with the wild type ATF6 α whereas the experiment in Figure 3 was carried out with the S1P cleavage site mutant of ATF6 α .
- The experiments from the first part of this paper show that, using a number of approaches, we can detect an interaction between ERp18 and ATF6 α and that this interaction increases upon ER stress. In addition, we show that ERp18 forms mixed disulfides with ATF6 α and that one of these mixed disulfides persists after ER stress. We go on to identify the specific cysteine residue within ATF6 α that is responsible for the formation of this mixed disulfide to ERp18. We feel that, taken together, Figs. 1-4 and Table 1 provide compelling evidence to support these conclusions. The point raised by the reviewer about the differences in interactions in the presence or absence of ER stress using different techniques is valid, however, the methods used to detect protein/protein interactions are distinct, which in some way explain the different results. The samples prepared for MS are first cross-linked and then immunoisolated with the final proteins eluted with a reducing agent. This ensures the removal of most non-specific binders, which is a necessary prerequisite due to the sensitivity of MS. The samples prepared for immunodetection on the other hand are crosslinked, immunoisolated but eluted with DTT and SDS. Finally, and in contrast, the mixed disulfides are a covalent interaction and show that ERp18 can reduce a disulfide in ATF6 and that this disulfide is stabilised with the substrate-trapping mutation. We have been careful not to overstate our conclusions following each approach and have changed the text accordingly. We have changed Fig. 3C as suggested.
- We would argue that the data shows a differential decrease in mixed disulfides to ERp18 (compare the mixed disulfide at ~150 kDa and at ~200 kDa). This differential decrease is in stark contrast to the mixed disulfides to the other PDIs including ERp57. The additional bands present in the PDIR blot (Fig 3D) are not present in the equivalent blots with anti-HA (Fig 3A, B, C lanes 5 and 6) indicating that they do not contain ATF6 α . PDIR seems to form non-covalent interactions with ATF6 as evidenced by the presence of endogenous PDIR in the ATF6 α immunisolate. The additional bands in Fig 3D may well be PDIR mixed disulfide to other secretory proteins.
- The purpose of Fig. 3D is to show that the mixed disulfide species contain both ERp18 and ATF6 α . We only included a thapsigargin treatment as the consequence of the different ER stressors is already shown in Fig. 3A, B, C.
- The ER stress treatment is for one hour so it is unlikely that the mixed disulfides between ATF6 α and ERp18 remain from before the stress treatment. However, unlikely this may be we have altered the text to indicate only that the mixed disulfides were present following ER stress.
- We base our suggestion (not absolute conclusion) on the fact that we could not find any other proteins interacting with the C618A mutant, which could explain the

much slower mobility of the oligomer compared to the C467A mutant. As there is only a single cysteine remaining in the luminal domain of these mutants they could only form disulfides to themselves or to other ER proteins. It seems sensible to suggest that the difference in mobility could be due to different conformations of the disulfide-bonded dimers that occupy different hydrodynamic volumes in a denaturing buffer.

- We have included the untreated controls for Fig. 5B as the reviewer suggested.
- We used DTT as an inducer of ER stress when assessing ER trafficking of ATF6 α due to the shorter time for activation with this reagent. We have explained this in the text and mentioned the fact that ERp18 delays ATF6 α trafficking even in the presence of DTT.
- This model is our speculation of the role of ERp18 during the activation of ATF6 α and while some aspects are clear, others are less so. The fate of the luminal domain of ATF6 α is unknown but it may be secreted. We have modified the model to make this clearer (it seemed to remain in the cytosol in the previous version). We do know that BiP dissociates from ATF6 α following ER stress and that ERp18 is recruited but the timing of these two events is not known. We have modified the text describing the model to make this clearer.
- The reviewer suggests an interesting experiment that we have now completed. Essentially, we determined the kinetics of transport of the single and double cysteine mutants of ATF6 after ER stress. The results show that there is no evidence for a more rapid trafficking of these ATF6 α mutants after ER stress. This result suggests that the retention caused by the presence of ERp18 is not solely due to the presence of disulfides. We have now incorporated this new data into figure 8.

Minor points:

- We have altered the first paragraph of the introduction to take into account the reviewer's suggestions.
- We now refer to Fig. 5G in the results section and include the effect of ERp18 expression in the KO cell line.
- The HDAC marker is only included in the nuclear fractions as it is a nuclear protein and would not be present in the membrane fractions. The relative amount of cleavage product seen in the membrane fractions derived from KO cells is clearly greater than that seen in the membranes from wild type cells indicating that its presence is not due to differential loading.
- The inhibitor was only included in Fig. 7A. The legend now states this clearly.

Reviewer #2

The reviewer had only minor queries to the text which we have been able to address.

Points outlined by reviewer.

- We have changed the sentence on page 6 to avoid confusion.
- The abbreviations have been added to the legend (Fig. 2A)
- The reviewer makes a valid point and we have changed the sentence on page 9 to reflect the possibility that a mixed disulfide may form by attacking a glutathionylated thiol.
- We have altered the text on page 10 to make our point clearer. In addition, we have modified Fig. 4C to depict our interpretation of the dimer mobility following non-reducing SDS-PAGE.
- We have included data now for the transcriptional regulation of ERp72 in Fig. 5.
- We define emPAI when first used
- We have altered Figure 1 to correct the problem with C618A. We have not changed the annotation as suggested by the reviewer as we think the listing of the cysteine mutants is quite clear when you compare with our annotation of the wild type protein. We include the R415A annotation, as there is no indication of the

position of R416 in the wild type. We think the figure quite clearly indicates the various mutations we have made.

- We have clarified the reason for the transfection of the RCN1 gene in the text.
- We have modified the figure legend to 7A accordingly.
- We have adjusted the use of commas throughout the manuscript as suggested.

Reviewer #3

#1. We note that the protein levels of BiP and ERp72 do not increase as much after ER stress in the ERp18 KO cells compared to the wild type. However, the attenuation at the mRNA level is most apparent with Grp78 whereas the relative levels of ERp72 are consistently but not significantly attenuated in the ERp18 KO cells. We have included this data in Fig. 5. We have not carried out a genome-wide transcriptional analysis on ER-stressed and unstressed cells as this is beyond the scope of the current work.

#2. The mobility of the ATF6 species shown on SDS-PAGE are indeed close together and are most apparent after we carry out the fractionation of membrane and nuclear extract. We base their identity on the disappearance of ATF6-N in the presence of the S1P inhibitor (Fig. 5E, lane 4) and the appearance of ATF6-M following ATF6 activation with thapsigargin (Fig. 6A, lane 2) or after brefeldin A treatment (Fig. 7B, lanes 2, 4, 6, 8). Both these species have faster mobility than ATF6-P. Under specific experimental condition we can observe the three species together which further confirms their identity. When the ERp18 KO cells expressing ATF6 are treated with brefeldin A we can observe the three products in the nuclear extract. We reproduce this figure below. Note the absence of ATF6-P in the wild type cells.

We consistently observe ATF6, ATF6-P and ATF6-M in the nuclear extract. The extract is prepared from a nuclear pellet that is resuspended in a buffer containing salts but in the absence of detergent. The resulting extract is centrifuged at 100,000 xg. It is possible that this extraction results in the release of some membrane from the nuclear material that is not pelleted during the final centrifugation. To test this possibility we also looked for the presence of another ER membrane protein (calnexin) in the nuclear extract. The results from this experiment are provided below and show that another ER membrane protein is present in the nuclear extract. This suggests that not all membrane material is pelleted during the final centrifugation step. The greatly diminished level of membrane-associated ATF6 in the nuclear extract after DTT treatment is likely due to the more efficient activation and therefore processing of ATF6 after this acute stress treatment.

#3. We discussed potential roles for ERp18 in regulating ATF6 trafficking, specifically in relation to its oxidoreductase activity and the requirement for disulfide reduction in the luminal domain of ATF6 prior to trafficking. The questions raised by the reviewer are ones that we are currently attempting to address but do not have any specific answers yet. The current paper demonstrates a role for ERp18 in regulating trafficking; the finer details of this process require further investigation.

#4. The question raised by the reviewer regarding a role for ERp18 in regulating the trafficking of SREBP is an interesting one which we have considered ourselves. While SREBP is cleaved in the Golgi by the same proteases as ATF6, regulation of its trafficking from the ER is likely to be different as it binds to the SREBP cleavage activating protein, which responds to levels of sterols. We investigated the trafficking of SREBP in our ERp18 KO cell line following incubation of cells in lipoprotein deficient serum containing media to induce Golgi trafficking and cleavage (lanes 3-6). We separated membrane and nuclear fractions and identified cleaved SREBP by the presence of an additional species in the

nuclear fraction (lanes 3 and 4, asterisk) that is absent when the experiment was carried out in the presence of the S1P inhibitor (lanes 5 and 6). Despite the high background, we observe equivalent cleavage of SREBP in the wild type and KO cells. This result suggests that ERp18 has little or no role in the regulation of SREBP trafficking.

#5. We also note the presence of ATF6-P in the nuclear fraction from ERp18 KO cells that diminishes following DTT treatment. It is more difficult to make the same conclusion regarding wild type cells as there is very little ATF6-P in this cell line (Fig. 6A, lanes 1-4). We provide a possible explanation for the presence of ATF6-P in the nuclear extract (see above). The fact that there is less present following DTT treatment could be due to the much more efficient trafficking and processing of ATF6 following DTT in comparison to thapsigargin treatment.

#6. The reviewer has correctly noted that ATF6-P is present in the ERp18KO cells transfected with ERp18. We see ATF6-P generated when we include the S1P inhibitor (Fig. 7A, lane 1,2) or express ATF6 with the S1P cleavage site mutation (Fig. 3). ATF6-P is also present in Fig. 8A (as indicated) albeit less intense than in Fig. 8C.

#7. The reviewer suggests that our results demonstrate that S1P is more effective in wild type cells than in the KO cells (Fig. 7B, lane 6 and 8). We are very cautious to draw this conclusion as the levels of expression of the ATF6 (D564G) mutant varies between these two cell lines. This figure is included only to show that S1P protease is still functional in the ERp18 KO cells.

Minor points:

#1. We have now changed figure 5 and include Fig. 5G in the results section.

#2. We have changed the text as indicated by the reviewer.

2nd Editorial Decision

24th April 2019

Thank you for submitting a revised version of your manuscript. We have now obtained new reviews from two of the original referees, which are copied below.

As you will see, they are satisfied with the revisions but also invite you to rephrase some misleading sentences in the text. In addition, before we can officially accept the manuscript, there are a few editorial issues concerning the text and the figures that I need you to address.

REFeree REPORTS

Referee #1:

In their revised manuscript, the authors have adequately addressed all my concerns and I now support publication in the EMBOJ. Only the sentence "The only protein which associated with ATF6 α after ER stress was the PDI family member ERp18" in the results should for sure be deleted/changed.

Referee #2:

The authors have attempted to address the concerns of this reviewer. Unfortunately, the clarifications were not made with great care in all cases, as the changes sometimes introduced additional problems. For example, the authors replaced a sentence with, "The interactions were specific to V5-tagged ATF6 α as the signals were much lower or not present in its absence." Now it seems from this sentence that it is something about the tagged version of ATF6 α that is causing the interaction! Which was presumably not the intention. The authors used the tagged version to pull out ATF6 α interactors, but they certainly hope those interactors are not specific for V5-tagged ATF6 α ! From Fig. 1C it is apparent that "absence" refers to absence of ATF6 α rather than absence of the V5 tag. But why confuse the reader in the main text?

It is also not clear why the authors removed the word "exclusively" to leave the false sentence, "The only protein which associated with ATF6 α after ER stress was the PDI family member ERp18." ERp18 is not the only protein to associate with ATF6 α after ER stress; the authors describe in the text and table 1 other proteins that associate with ATF6 α after ER stress. The authors presumably meant to describe ERp18 as a protein that was not seen to associate with ATF6 α in the absence of ER stress but was seen to associate following ER stress. That is a completely different statement.

Also unfortunately, the authors did not reproduce the reviewers' comments in their point-by-point response, only the responses themselves. Consequently, it is somewhat difficult to determine whether the points of the other reviewers were adequately addressed.

Nevertheless, this reviewer maintains the opinion that the results are an important advance and that the finding that ATF6 α is trafficked to the Golgi but processed aberrantly in the absence of ERp18 is interesting.

Corresponding Author Name: Prof. Neil Bulleid

Journal Submitted to: The EMBO Journal

Manuscript Number: EMBOJ-2018-100990